# Potential faster Arctic sea ice retreat triggered by snowflakes' greenhouse effect

J-L F. Li[1], Mark Richardson[1,2], Wei-Liang Lee[4], Eric Fetzer[1], Graeme Stephens[1], Jonathan Jiang[1]
Yulan Hong[3], Yi-Hui Wang[6], Jia-Yuh Yu[7], Yinghui Liu[5]

[1]Jet Propulsion Laboratory, California Institute of Technology, Pasadena, CA 91125, USA
[2]Joint Institute for Regional Earth System Science and Engineering, University of California, Los Angeles, CA 90095-7228 USA
[3]Department of Earth, Ocean and Atmospheric Science, Florida State University, Tallahassee, FL 32304, USA
[4]RCEC, Academia Sinica, Taiwan
[5]Cooperative Institute for Meteorological Satellite Studies, University of Wisconsin, Madison, WI 53706, USA
[6]Center for Coastal Marine Sciences, California Polytechnic State University, San Luis Obispo, CA 93407, USA
[7]Department of Atmospheric Sciences, National Central University, Taoyuan City, 32001, Taiwan

*Correspondence to*: J-L Frank Li (Juilin.F.Li@jpl.nasa.gov)

**Abstract.** Recent Arctic sea ice retreat has been quicker than in most general circulation model (GCM) simulations. Internal variability may have amplified the observed retreat in recent years, but reliable attribution and projection requires accurate representation of relevant physics. Most current GCMs don't

fully represent falling ice radiative effects (FIRE), and here we show that the small set of Coupled Model Intercomparison Project, phase 5 (CMIP5) models that include FIRE tend to show faster observed retreat. We investigate this using controlled simulations with the CESM1-CAM5 model. Under 1pctCO2 simulations, including FIRE results in the first occurrence of an "ice free" Arctic (monthly mean extent $< 1 \times 10^6$ km$^2$) at 550 ppm $CO_2$, compared with 680 ppm otherwise. Over 60—90 °N oceans, snowflakes

reduce downward surface shortwave radiation and increase downward surface longwave radiation, improving agreement with the satellite-based CERES-EBAF surface dataset. We propose that snowflakes' equivalent greenhouse effect reduces the mean sea ice thickness, resulting in a thinner pack whose retreat is more easily triggered by global warming. This is supported by the CESM1-CAM5 surface fluxes and a reduced initial thickness in perennial sea ice regions by approximately 0.3 m when FIRE are

included. This explanation does not apply across the CMIP5 ensemble where inter-model variation in the simulation of other processes likely dominates. Regardless, we show that FIRE can substantially change Arctic sea ice projections and propose that better including falling ice radiative effects in models is a high priority.

## 1 Introduction

The Arctic region is undergoing pronounced change, becoming warmer and wetter (Boisvert and Stroeve, 2015) while its land ice melts (Jacob et al., 2012; Kjeldsen et al., 2015) and spring arrives weeks earlier than in the 1990s (Post et al., 2018). Communities in the region may have to adapt to changing hunting seasons (Rolph et al., 2018), loss of coast that was previously protected by sea ice (Overeem et al., 2011) and surface destabilisation due to permafrost melt (Shiklomanov et al., 2017).

In particular, Arctic sea ice retreat potentially opens area for resource extraction or transport routes (Smith and Stephenson, 2013) and has national security implications for neighbouring states. Physically, sea ice affects both top-of-atmosphere and surface heat fluxes. In winter it insulates the ocean, restricting the leakage of heat to space via infrared cooling, and in summer it predominantly reflects sunlight and cools the surface (Tietsche et al., 2011). Throughout the year it restricts evaporation and therefore affects the hydrological cycle (Bintanja and Selten, 2014). It has been proposed that reduced sea ice extent may further smooth the latitudinal temperature gradient, thus weakening the high latitude jets and making it easier to shift into a "wavy" pattern, which is associated with long-lived extreme events at mid latitudes (Francis and Vavrus, 2012). However, these proposed impacts at lower latitudes are currently speculative and disputed (Cohen et al., 2014).

The recent rapid Arctic sea ice retreat included extreme minima in 2007 and 2012 which received particular attention. Regarding the 2007 minimum, a reduction in cloudiness during the melt season relative to previous years was shown to change surface energy balance by enough to thin sea ice by up to 0.3 m over three months (Kay et al., 2008). Atmosphere and ocean dynamics may also export ice to lower latitudes. For example, stronger circulation associated with the Arctic Oscillation can increase the total area of new, thin ice but transport the thicker ice away from the coldest regions and leave it vulnerable to

summer melting (Rigor et al., 2002). Surface pressure observations have been used to infer contributions to summer sea ice reduction due to anomalously high ice export through the Pacific sector in 2007 (Zhang et al., 2008) and the Fram Strait in 2012 (Smedsrud et al., 2017).

Based on CMIP5 output (Climate Model Intercomparison Project, phase 5; Taylor et al. (2012)), the
observed extreme low events and general retreating trend have been attributed to a combination of melt driven by global warming along with  internal variability, such as extreme cloud anomalies affecting surface radiation (Kay et al., 2011) and from 1990 through the early 2000s, potentially wind-driven factors (Rigor and Wallace, 2004). One recent study suggested an equally important role for anthropogenic warming and natural variability for the extreme 2012 loss (Kirchmeier-Young et al., 2017).

Reliable attribution requires the ability to quantify physical processes and relevant responses to each forcing. A better understanding of the processes that are responsible for sea ice retreat will help to reduce uncertainties in future projections. Accurate future projections are necessary for informed decisions with the changing Arctic, such as by investors or insurance companies who may wish to assess the risk associated with proposals for future shipping routes. A common criterion is determining if and when a
seasonally "ice free" Arctic will occur, arbitrarily defined as when sea ice extent falls below $1 \times 10^6$ km$^2$. At this point the remaining ice would cluster around islands and coasts, leaving the basin largely open. Climate models are crucial tools to inform projections but their Arctic response varies widely (Massonnet et al., 2012; Stroeve et al., 2012). The time at which the Arctic is likely to become "ice free" under high radiative forcing in CMIP5, for example, ranged from 2041—2060 in Massonet et al. (2012) while
Stroeve et al. (2012) only stated that "a seasonally ice-free Arctic Ocean within the next few decades is a distinct possibility".

Observed summer retreat has been faster than the average CMIP5 model simulation and if the CMIP5 models do not adequately include factors that influence sea ice retreat then their projections will be biased. We have previously shown that the majority of CMIP5 models do not properly account for atmospheric
ice in their radiation codes. While they include suspended ice, falling ice is excluded and this causes region-dependent biases in the surface energy budget that, for example, tend to result in a larger mean Antarctic sea ice extent (Li et al., 2017).

Here we focus on sea ice extent changes and the surface energy budget over oceans from 60—90 °N. In the simplest terms, falling ice should produce a year round increase in downward surface longwave radiation ($LW_\downarrow$) and a decrease in downward surface shortwave radiation ($SW_\downarrow$), which will be greatest in local summer. Li et al. (2017) showed that in the Antarctic this results in a dampened annual cycle with the increased wintertime $LW_\downarrow$ restricting maximum sea ice extent, which then results in a lower albedo when the sun rises again. This lower albedo counteracts somewhat the reduction in sunlight arriving at the surface due to reflection by snowflakes.

With regards to the Arctic, we expect a somewhat different response due to (1) wintertime maximum extent being restricted by continental boundaries and boundaries with warm ocean currents, (2) generally thicker sea ice (Kurtz and Markus, 2012; Kwok and Cunningham, 2008) and (3) faster local warming under the early part of $CO_2$-driven heating.

It is therefore possible that increased $LW_\downarrow$ from FIRE may not have a substantial effect on winter sea ice extent, but may restrict its thickness. This should favour faster retreat in sea ice cover both during a typical summer melt season and during long-term warming. However, if the maximum wintertime extent is not strongly affected then the albedo will begin the melt season at a similar level regardless of FIRE and a non-FIRE simulation should have a stronger local sea ice albedo feedback due to its stronger $SW_\downarrow$. The $SW_\downarrow$ and $LW_\downarrow$ effects from including FIRE should oppose each other and it is not necessarily obvious whether one factor will dominate.

From the Antarctic sea ice results of Li et al. (2017), we suspect that the longwave effect is more important for the mean state. Our hypothesis is that FIRE increases year round $LW_\downarrow$ and results in a thinner sea ice cover on average. It is then easier to melt this pack as temperatures warm and our hypothesis is related to the recent findings of Massonnet et al. (2018) who also describe several relevant physical processes. They found that across CMIP5 models, sea ice retreat is correlated with parameters representing seasonal growth and retreat. They considered differences between the level of sophistication of the sea ice components of the CMIP5 models and found that the background thickness was more strongly related to sea ice retreat than model sophistication. This sensitivity of sea ice retreat to initial thickness supports our

hypothesis although we focus on an atmospheric driver of changes in the initial mean state of thickness, namely FIRE.

As well as changes in the mean state which could affect retreat through the initial pack's robustness, it is also possible that local fluxes could vary in different ways under warming. For example, in a simulation where FIRE are included, warming could raise the atmospheric melting layer during summer, leading to a reduction in snow water path in favour of rainfall, which is not included in the radiation code. The direct consequence of this would be to reduce the trend in $LW_\downarrow$ and increase the trend in $SW_\downarrow$, relative to a simulation where FIRE are excluded. This ignores further coupling to atmospheric conditions that could similarly affect feedbacks.

Here we investigate the importance of FIRE using both standard CMIP5 output along with simulations with a CMIP5 era climate model, the National Center for Atmospheric Research-Department of Energy (NCAR-DOE) Coupled Earth System Model version 1 with the Coupled Atmosphere Model version 5 (CESM1-CAM5). We refer to these as our "controlled" simulations to emphasise that we controlled the inclusion of FIRE and to distinguish them from other studies' CESM1 simulations.

Our two main aims are to determine whether FIRE substantially change simulated Arctic sea ice and, more specifically, to test our hypothesis that FIRE tends to reduce mean initial sea ice thickness and thereby leave it more vulnerable to retreat under warming.

CMIP5 output will be used to determine whether differences in simulated sea ice can be detected between FIRE and non-FIRE models across the ensemble, and if so whether the changes can be linked to radiative heat fluxes in a way consistent with our expectations from FIRE.

The CMIP5 models have many differences that may affect sea ice extent, most obviously in their sea ice components. The sea ice albedo schemes for example vary in their sophistication and treatment of snow on ice, melt bonds and response to temperature. The resultant inter-model spread in local albedo feedback does not appear to explain much of the inter-model variance in long-term retreat (Koenigk et al., 2014), but modelled sea ice albedo does correlate with the amplitude of the annual cycle sea extent (Karlsson and Svensson, 2013). As described in Massonnet et al. (2018), any process that affects the baseline thickness may be related to future retreat, and this includes ocean eddy heat flux (Horvat and Tziperman,

2018) and cloud schemes that affect surface radiation and temperature change. For example in CESM1-CAM5.1, matching the the observed prevalence of mixed phase clouds at low temperatures (Cesana et al., 2012, 2015) results in approximately 1 °C more warming under $CO_2$ doubling (Tan et al., 2016). Such a large increase in warming would be expected to also change projected sea ice extent. Differences in sea

ice, ocean and atmosphere schemes may drive changes that confound detection of FIRE-driven sea ice effects across the CMIP5 ensemble so our analysis of controlled CESM1-CAM5 simulations in which the only difference is the inclusion of FIRE allows a direct comparison. In these simulations our analysis ignores coupled dynamical responses in favour of studying the surface radiative flux terms that provide a direct test of our hypothesis. The paper is structured as follows: Section 2 lists the data and methodology,

Section 3 reports on the simulated and observed sea ice changes, Section 4 looks at the simulated and observed surface radiative fluxes, Section 5 synthesises and discusses the results and their limitations, and Section 6 concludes.

## 2 Methods and Data

### 2.1 CMIP5 and CESM1-CAM5 Simulations

We use outputs from the CMIP5 archive (Taylor et al., 2012) and select models that provide all surface energy balance terms plus the fields necessary to calculate sea ice extent for the preindustrial control (piControl), historical and Representative Concentration Pathway 8.5 (RCP8.5, Riahi et al. (2011)) scenarios. The historical scenarios run through 2005, after which we append the RCP8.5 output. This is a scenario of very high radiative forcing, which we select to better identify forced response over internal

variability, and we make no judgment about the probability that this forcing will occur. For each model we select the first simulation in each case, r1i1p1 in CMIP5 nomenclature, which results in 25 simulations. We split these into two sub-ensembles depending on whether FIRE are allowed: those including snow radiative effects (CMIP5-SoN, N = 7) and those in which falling snow radiative effects are not considered (CMIP5-NoS, N = 18). All models are listed in Supplementary Table 1.

For CESM1-CAM5 we use previously published historical simulations (Li et al., 2014), which are run on a spatial resolution close to a 1×1° latitude-longitude grid and follow the CMIP5 historical protocol.

CAM5 is one of the few atmospheric models that allows snow radiative interactions, and it does this thanks to a two-moment treatment of rain and snow. Falling snow mass and the crystal number concentration is diagnosed at each model level and time step, and is related to an effective radius as detailed in Section 2 of Morrison and Gettelman (2008). The profile of snow mass and effective radius is then related to radiative properties using precomputed lookup tables based on an assumed ice habit mixture as described in Section 2.5 of Gettelman et al. (2010). The scheme only represents the stratiform component of falling ice and not that in convective towers, but the majority of Arctic snowfall will be included. With this scheme snow radiative effects can be allowed (CESM1-SoN) or disallowed (CESM1-NoS), and the inclusion or exclusion of FIRE is the only difference between the SoN and NoS simulations. The radiative effects of rain are not included in any of the CESM1-CAM5 simulations, but this is unlikely to be an issue for much of the Arctic. Even ignoring the differences in how rain and snow affect radiation, CloudSat radar-based products report that Arctic precipitation frequency and amount is dominated by snow (Behrangi et al., 2016).

The strength of FIRE and the simulated response of other properties to FIRE depend on the frequency as well as the intensity of snowfall. This is accounted for in the model as radiative transfer is calculated at each model time step even though outputs are only provided monthly. Note that the CESM1-SoN and CESM1-NoS simulations are independent so will have different amounts and patterns of snowfall, and that by including FIRE there can be coupled changes in heating rates, circulation and precipitation (Chen et al., 2018). We later use the SoN-NoS surface radiative flux differences because these include the full coupled changes due to FIRE and are the properties most directly relevant for sea ice changes.

Unfortunately, output is not available for any RCP, which forces observational comparisons to end in 2005. To estimate how sea ice extent changes under greater forcing we use output from available simulations following the CMIP5 1pctCO2 protocol in which atmospheric $CO_2$ increases at 1 % $yr^{-1}$ for 140 years from an initial value near 280 ppm. Radiative forcing estimates differ, but typical values for quadrupled $CO_2$ are 5.3—8.6 W $m^{-2}$ (Forster et al., 2013) , meaning that total forcing is similar to the historical-RCP8.5 series used for CMIP5. We use output from fully coupled CESM1-SoN and for CESM1-NoS runs following the historical and 1pctCO2 simulations.

## 2.2 Sea Ice Extent

Sea ice extent (SIE) is defined as the area of ocean with sea ice concentration (sic) greater than 15 %. This was originally developed for satellite-based passive microwave products to be a robust identifier of ice edges when compared against aircraft observations (Cavalieri et al., 1991). This threshold means retrieved sea ice edges are less sensitive to changing weather conditions or melt ponds on the ice which may interfere with the observed brightness temperatures. For observations we use the National Snow and Ice Data Center (NSIDC) monthly series of total sea ice extent (Fetterer et al., 2017) which is calculated from gridded data on a nominal 25 km grid. We use the complete years that were available as of analysis time: 1979—2017.

The standard CMIP5 output is the sea ice concentration within an ocean grid cell, and we calculate sea ice extent following a previously published method (Kirchmeier-Young et al., 2017), by reporting the total area of all of the model's native ocean grid cells with sic > 15 % (see Supplementary Figure 1 for verification of this calculation). This is not a fully consistent comparison due to differences in grid cell sizes and as observations may underestimate sea ice concentration in the presence of substantial melt ponds. Here we assume that these factors have little effect on the large-scale changes under study.

To represent the magnitude of changes in SIE we apply optimised least squares (OLS) to each calendar month's time series separately (e.g. all Januaries for 1979—2005) assuming Gaussian white noise and report both trend estimates and their associated errors. We justify this based on analysis of the detrended residuals of the NSIDC dataset applied to 1979—2005 and 1979—2017. While some months reject white noise at $p < 0.05$ according to the Ljung-Box test applied for lag-1 autocorrelation, these results are not robust since no calendar month rejects white noise over both periods. No month shows residuals that are significantly different from normality according to the Kolmogorov-Smirnov test: see Supplementary Table 2 for summary of Pearson's $r$, Ljung-Box $p$ and Kolmogorov-Smirnov $p$.

## 2.3 Sea Ice Thickness

Given that our hypothesis is that FIRE drives changes in the initial mean sea ice thickness, we also compare the CESM1-SoN and CESM1-NoS sea ice thickness in the 1pctCO2 simulations. Regional

average sea ice thickness is calculated by appropriately area weighting the ice covered area of each grid cell included in the region. For a consistent comparison we select all grid cells where both simulations have greater than 80 % mean sea ice concentration for all calendar months averaged over years 1—20 and 21—40 of their 1pctCO2 simulations. The selected region changes between each period, and a static region poleward of 80 °N as in Massonnet et al. (2018) is also shown. The 80 % concentration threshold means the areas are consistently ice covered and includes about five times as much area as using a 90 % threshold, so our thicknesses are more representative than using a stricter cut off (Supplementary Figure 2). The mean thickness in each region is calculated for each calendar month and our hypothesis is supported if the CESM1-SoN mean thickness is greater than the CESM1-NoS mean thickness in this region.

## 2.4 Surface Energy Budget

We use $1°×1°$ monthly estimates of surface fluxes from the Clouds and the Earth's Radiant Energy System Energy Balanced and Filled-Surface (CERES-EBAF Surface, Kato et al. (2013)) product, for which we have complete years for 2001—2015. CERES-EBAF Surface combines satellite data with a radiative transfer model to calculate surface fluxes and is estimated to have a monthly root mean square error of $±11$ W m$^{-2}$ in each surface radiative flux term over oceans (Kato et al., 2012).

CESM1-CAM5 output is provided monthly at $1°×1°$ and for all CMIP5 models, we use previously interpolated $2.5°×2.5°$ monthly data. Fluxes are calculated by taking the area-weighted average of values in each grid cell after scaling by the ocean fraction (total ocean fraction, including sea ice covered ocean). For CERES and CESM1-CAM5 we use the CESM1-CAM5 land sea mask, and for all CMIP5 models we use a consistent fractional land sea mask built from the $0.125°×0.125°$ European Center for Medium Range Weather Forecasts European Reanalysis-Interim (ECMWF ERA-Interim) land mask. For comparison of the mean state fluxes between CERES and our controlled historical CESM1-CAM5 simulations we only have 5 complete years of overlap, 2001—2005 inclusive.

We consider the difference CESM1-CAM5 minus CERES but since our simulations are coupled, internal variability could increase the apparent model-observation discrepancy. As an estimate of the magnitude

of internal variability on our 5-year averaged fluxes, we detrend the model output over 1981—2005, and the CERES output over 2001—2015 then slice these into non-overlapping 5 year periods. The standard deviation is calculated for the modelled and observation-based samples and then these are added in quadrature to provide a value for the CESM1-CAM5 minus CERES difference. This estimate only represents the effect of internal variability due to our use of a short time period, and may be biased if the variance in these terms changed greatly from 1979—2015. Given the brevity of the available data record we consider this simple approach to be adequate.

## 3 Observed and Simulated Sea Ice Extent and Thickness Results.

### 3.1 Sea Ice Extent

Figure 1 shows the March and September post-1979 SIE in NSIDC observations and CMIP5 simulations. These are the months of maximum and minimum SIE (all months are shown in Supplementary Figure 3). Figure 1(b) shows that observed September retreat approaches the lower 10[th] percentile of the CMIP5 ensemble. When plotted using anomalies, the retreat falls outside the model range (see Supplementary Figure 4 for absolute anomalies, Supplementary Figure 5 for relative anomalies).

In Figure 1(c,d) the results are split into CMIP5-SoN and CMIP5-NoS sub-ensembles, with Figure 1(d) showing that CMIP5-SoN better captures the observed September retreat over 1979—2017. The median CMIP5-SoN trend is more negative than that of CMIP5-NoS from June through October, in better agreement with observations (Supplementary Figure 6). In March, trends are similar but CMIP5-SoN shows greater extent, which is the opposite of expectations if wintertime $LW_\downarrow$ from FIRE were the main cause of differences. However, inter-model differences in parameterisations and calculation methods for the atmosphere, oceans and sea ice can change the mean state, so to isolate FIRE we present the controlled CESM1-CAM5 simulations in Figure 2.

Over 1979—2005 there is a smaller discrepancy between CESM1-CAM5 and observations for monthly mean extent when including FIRE (see Supplementary Figure 7). Retreat during the same period is faster in CESM1-SoN than in CESM1-NoS: for September the SoN minus NoS series is significant at $2.61\sigma$ (white noise $p = 0.01$, see Supplementary Figure 8). The CESM1-SoN September retreat is faster than in

reality over 1979—2005, but not significantly so ($p = 0.06$). Real world Arctic sea ice retreated more rapidly after 2005, but we do not have the output to determine whether this means that CESM1-SoN would then show better agreement. For increased warming we must turn to the 1pctCO2 output, and Figure 2(d) shows accelerated retreat in CESM1-SoN following year 40, corresponding to $CO_2$ levels of

416 ppm, a value that current trends suggest will occur in the 2020s.

To allow easier interpretation, we take overlapping decadal averages of mean SIE and the number of years within that decade with SIE $< 1 \times 10^6$ km$^2$, and plot these as a function of atmospheric $CO_2$ concentration (year 0 is approximately 280 ppm) in Figure 3. Below the 2017 atmospheric $CO_2$ concentration, Figure 3(a) shows only small differences in decadal mean September SIE, but for concentrations higher than this

the Arctic sea ice retreats far more rapidly under global warming when FIRE are included. Note that these simulations exclude non-$CO_2$ forcings such as aerosol, which are present in reality. In the CESM1-SoN simulation, Figure 3(b) shows that the majority of years are classified as ice free once atmospheric $CO_2$ passes 550 ppm, compared with 680 ppm in the CESM1-NoS simulation. In a naïve sense (i.e. assuming approximately constant airborne fraction as occurs for these decades in some 1pctCO2 simulations, e.g.

Matthews et al. (2009)) this implies a difference of almost 100 % in cumulative future anthropogenic $CO_2$ emissions before the Arctic commonly becomes ice free if these CESM1-CAM5 1pctCO2 simulations are representative of the real world. Figure 3 shows that the potential impact of FIRE on Arctic sea ice retreat is large, but we do not argue that this necessarily means that Arctic sea ice will necessarily collapse more rapidly than indicated by CMIP5. Firstly, CESM1-CAM5 may have compensating biases due to

other processes and secondly the disappearance of ice under transient $CO_2$-driven warming may not correspond to reality where a mixture of radiative forcing agents is changing. Some of these, such as aerosols, may drive stronger seasonal, regional, and dynamic responses than well-mixed greenhouse gases like $CO_2$ (Hansen et al., 1997).

A further consideration is that internal variability can change when an ice free state occurs. Under RCP8.5

the CESM1 large ensemble of 40 runs (Kay et al., 2015) shows a 14-year range between members when ice free is defined based on the five-year average (Jahn et al., 2016). The CESM1-SoN to CESM1-NoS

1pctCO2 difference by this criterion is 20 years so our conclusion that FIRE drives faster retreat is likely robust to internal variability.

These simulations show that falling ice radiative effects could lead to much greater Arctic sea ice retreat when the system is forced under global warming and support the inclusion of FIRE in future modelling efforts. Next, we investigate whether the surface radiative energy balance allows us to identify candidate physical processes that explain these changes, and whether the processes identified using CESM1-CAM5 can be detected across the CMIP5 ensemble.

## 3.2 Sea Ice Thickness in CESM1-SoN and CESM1-NoS

Figure 4(a,d) outline the regions within which thickness is calculated for years 1—20 and 21—40, and the annual cycles of mean thickness for each period and simulation are shown in Figure 4(b,c,e,f). Consistent with our hypothesis, the CESM1-SoN ice pack starts off thinner than that of CECSM1-NoS. Over the Arctic interior the pack tends to be 20—30 cm thinner throughout the year. The remaining perennial >80 % sea ice concentration region for years 21—40 in Figure 4(e) shows a 1.4 m difference.

## 4 Observed and Simulated Surface Radiative Fluxes

## 4.1 CESM1-CAM5 Controlled Simulations

In Section 1 we discussed the expected direct effects of FIRE on surface longwave (LW) and shortwave (SW) radiative fluxes and how these might be related to SIE. We begin our analysis with the downward fluxes at the surface, $LW_\downarrow$ and $SW_\downarrow$ in CESM1-CAM5 compared with CERES-EBAF Surface observations during their overlap period of 2001—2005. Uncertainties are based on the standard deviation of non-overlapping five-year periods from the rest of their records as described in Section 2. The CESM1-CAM5 minus CERES-EBAF Surface flux differences over 60—90 °N oceans are displayed in Figure 5 for each calendar month. As expected, inclusion of FIRE results in increased $LW_\downarrow$ and decreased $SW_\downarrow$, resulting in better agreement with the observation-based CERES data. Figure 5(a) shows that the $LW_\downarrow$ difference is greatest in winter, when the $SW_\downarrow$ is negligible due to the lack of available sunlight. The SoN-NoS difference in $SW_\downarrow$ is greater than in $LW_\downarrow$ during summer, but only marginally so, and the annual

average LW$_\downarrow$ difference is greater. From Figure 5(b), the net absorbed surface SW radiation shows relatively small SoN-NoS differences because while FIRE reduces SW$_\downarrow$, it also reduces SIE and so lowers the mean albedo. The net absorbed surface longwave radiation is consistently greater in SoN, explaining the majority of the remaining difference in net radiation in Figure 5(c). The annual average annual average

LW$_\downarrow$ is 11 W m$^{-2}$ higher when including FIRE, which will increase mean ice temperature and increase heat input, resulting in a thinner pack that is more vulnerable to warming.

It is also possible that local radiative feedbacks could be different when including or excluding FIRE. This would manifest as a change in the SoN minus NoS flux differences over time and for this we switch back to analysis of the 1pctCO2 simulations. Figure 6 includes the 1pctCO2 SW$_\downarrow$ and LW$_\downarrow$ differences

for each season: December-January-February (DJF), March-April-May (MAM), June-July-August (JJA) and September-October-November (SON). Long-term trends are estimated by multiplying the OLS trend gradient by the length of the period, and the only significant ($p < 0.05$) trend occurs in SON where there is a decrease in the radiative flux difference between the two simulations.

However, the SoN minus NoS LW$_\downarrow$ trend is insignificantly positive during the first 70 years (+0.08±0.09

W m$^{-2}$ yr$^{-1}$, ±2σ error in OLS trend), so the full-period LW$_\downarrow$ trend is not responsible for driving the faster disappearance of sea ice in CESM1-SoN which has largely occurred by year 70 (as shown by Figure 2(d) for September). Instead, the difference appears related to differences in the relative effects of FIRE between icy and ice-free states. During the first 40 years when the simulations both have a healthy Arctic ice cover the mean SON difference in LW$_\downarrow$ is 11.6±11.1 W m$^{-2}$ (±2 standard deviations) whereas for the

final 40 years where both simulations are ice free during September, the difference is 7.1±6.6 W m$^{-2}$. This difference could be related to changes in cloud properties or phase in response to sea ice cover; the CESM1-SoN simulation initially has a smaller sea ice extent but by the end of the simulation both CESM1-SoN and CESM1-NoS are largely ice free.

Taken together, the energy budget analysis of CESM1-CAM5 1pctCO2 simulations indicates that

differences in flux trends due to FIRE do not drive the faster observed retreat, but instead the effect of stronger year-round LW$_\downarrow$ in the initial state is the most important radiative contribution. This supports our argument that the effective greenhouse effect from snowflakes results in a thinner pack whose retreat is

more easily triggered by warming. This snowflake greenhouse effect is present year round and throughout the entire Arctic basin, leaving no safe spaces where the ice can fully recover.

## 4.2 CMIP5 Ensemble Results

The CESM1-CAM5 results show that snow radiative effects can substantially change simulated Arctic sea ice retreat under warming, which is consistent with the generally earlier disappearance of sea ice seen under historical-RCP8.5 simulations for the CMIP5-SoN sub-ensemble, compared with the CMIP5-NoS ensemble. To investigate this, we consider the CMIP5 1979—2005 mean annual cycle and the 2006—2035 linear regression trends for each calendar month for a variety of properties in Figure 7. Each simulation's line is coloured according to whether it includes FIRE (SoN, blue) or excludes FIRE (NoS, red).

The mean state period is the overlap between NSIDC passive microwave sea ice extent data and the historical simulations, and the trend period covers 30 years in which Figure 1 shows an apparent notable divergence in SIE between the CMIP5-SoN and CMIP5-NoS sub-ensembles.

Inspection of Figure 7 shows no clear support across the CMIP5 ensemble for the hypothesis we developed using the controlled CESM1-CAM5 simulations. In fact, Figure 7(d) shows that two models that include FIRE show substantially more summertime $SW_↓$ (e.g. 45 W m$^{-2}$ more than the median of all other CMIP5 models), which is the opposite of the direct effects we hypothesise are related to FIRE. These models are GISS-E2-H and GISS-E2-R, whose CMIP5 versions greatly underestimated mean Ice Water Path (IWP) poleward of 60—90 °N (Stanfield et al., 2014). This illustrates how other differences aside from FIRE may well have compensating effects, showing that FIRE alone is insufficient to explain differences in Arctic sea ice retreat among models.

## 5 Discussion and Conclusions

The apparent agreement in September sea ice retreat between CMIP5-SoN and CESM1-SoN seen in Figure 1 and Figure 2 appears supportive of a major role for falling ice radiative effects in reinforcing Arctic sea ice retreat. However, the CMIP5 result was largely due to extremely early ice disappearance

in the GISS-E2 models which accounted for two out of seven of the sub-ensemble members. These models have been shown to drastically underestimate total ice water path, resulting in too much SW↓ during summer and therefore likely a very strong surface albedo feedback. As detailed in Section 1, simulated sea ice is affected by many model design factors including the sea ice albedo scheme, ocean eddy heat transport and cloud simulations. Therefore, the CMIP5 cross comparison simply shows that Arctic sea ice projections are at least as sensitive to other factors as to the inclusion or exclusion of FIRE and the faster September retreat of CMIP5-SoN in Figure 1 is likely due to the full combination of properties in these models and not directly due to FIRE. Nevertheless, the controlled CESM1-CAM5 simulations demonstrate that the inclusion of FIRE in this model results in a thinner sea ice pack and a faster retreat in extent over both 1979—2005 and in 1pctCO2 simulations. The difference between CESM1-SoN and CESM1-NoS in 1pctCO2 is larger than the range of values due to internal variability spanned by the CESM1 large ensemble so we conclude that we have detected a FIRE-driven difference in modelled Arctic sea ice retreat.

While we did not explore any dynamic changes in response to the inclusion of FIRE. The magnitude of the radiative effects are a credible candidate for explaining major differences in sea ice extent, with 11 W m$^{-2}$ of downward longwave radiation over a year being sufficient to melt ~1 metre of ice annually assuming that all of the heat goes into the ice (Kay et al., 2008). In reality this thinning is reduced by negative feedbacks, and Figure 4(b,c) show that in CESM1-CAM5 the net result is a thinning of approximately 30 cm of the interior ice pack, consistent with our hypothesis that FIRE thins the Arctic ice pack and preconditions it for more rapid melt. Nevertheless, changes in dynamics that affect patterns of cloudiness, ice transport or ocean heat transport could reinforce or counteract our proposed changes and we have not investigated these.

In conclusion, we do not argue that the exclusion of FIRE in current models necessarily means that Arctic sea ice will retreat faster than simulated by the average CMIP5 model. CESM1-CAM5 might show a stronger sea ice response to FIRE than other models or, following inclusion of FIRE that, modellers might tune other processes in a way which counteracts FIRE-driven sea ice changes. Or a model may have a stronger summertime albedo feedback than longwave radiation-driven thinning effect, and show slower

retreat once FIRE are included. However, our controlled experiments show a strong sensitivity of sea ice projections to FIRE in at least one model, with Figure 2(d) showing September Arctic sea ice retreat being approximately twice as fast once atmospheric $CO_2$ concentrations are above 2017 levels under 1 % $yr^{-1}$ CO2 growth. Given that the snow radiative effect exists in reality, we encourage other modelling groups

to include them in future cloud schemes to increase confidence in Arctic sea ice projections.

*Data availability.* The NSIDC
[ftp://sidads.colorado.edu/DATASETS/NOAA/G02135/north/monthly/data/], CERES-EBAF [https://ceres.larc.nasa.gov/order_data.php] and CMIP5 data [https://cmip.llnl.gov/cmip5/data_portal.html] are available from public archives. The time series of CMIP5 and CESM1 sea ice and radiative fluxes are appended as supplementary information.

*Author contributions.* JLL led the research and performed the CESM1 sensitivity output processing and analysis. WLL conducted CESM1 model sensitivity runs. MR performed the CMIP5 processing and the time series analysis. YHW, YLH and JYY provided discussion and editing. EF, JJ and GS supported and offered comments and suggestions to the study. YL quality controlled the data.

*Competing interests.* The authors declare no competing interests.

*Acknowledgements.* MR thanks Dr. Kirchmeier-Young for providing her CMIP5 sea ice extent series to allow verification of his code.

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

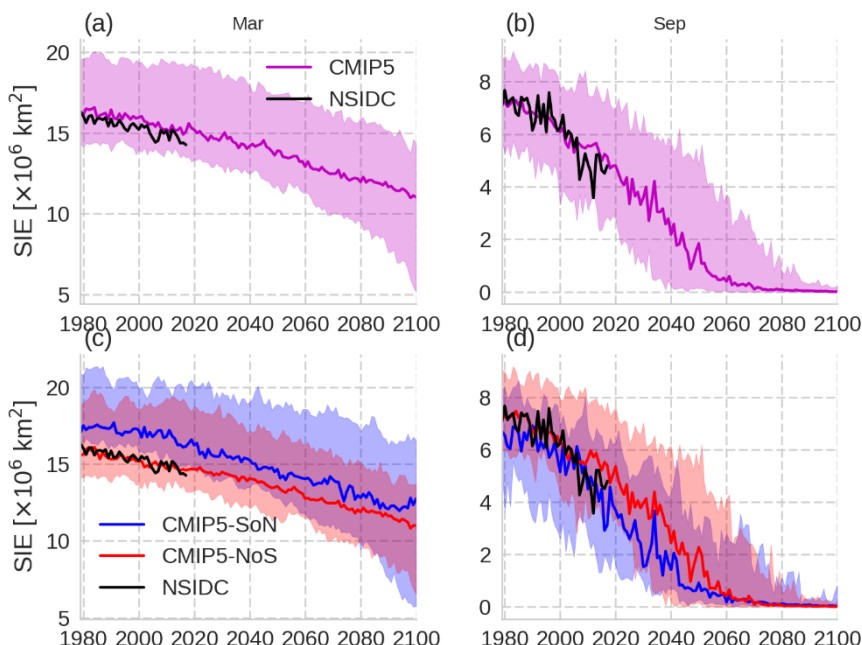

**Figure 1: Arctic sea ice extent in NSIDC observations (black) and CMIP5 climate models (line is ensemble median, shading is 10—90 % range). (a) full ensemble in March, (b) full ensemble in September, (c) CMIP5 split into sub-ensembles of models with FIRE (CMIP5-SoN) and those without (CMIP5-NoS) in March and (d) SoN and NoS in September.**

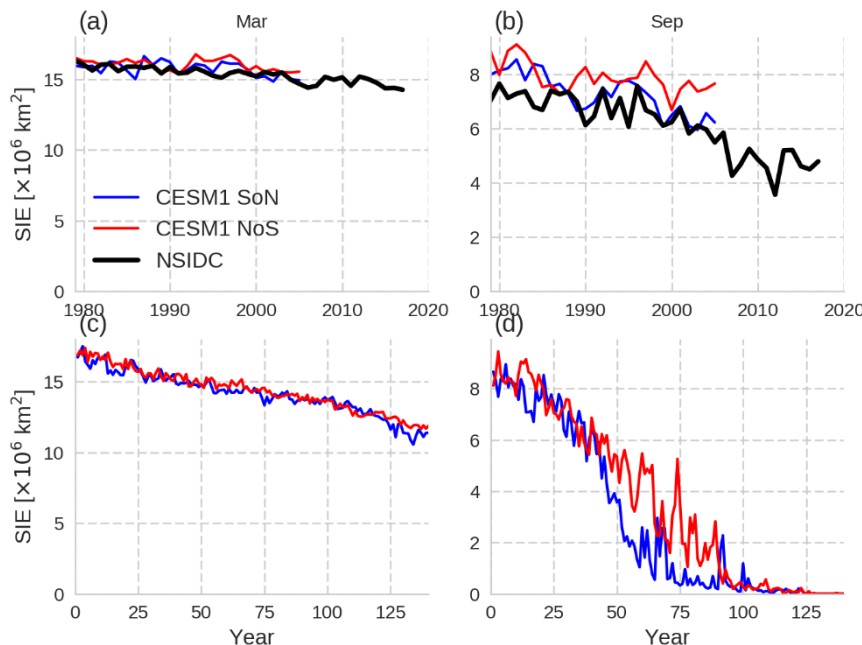

**Figure 2: Observed (black) CESM1-CAM5 simulated Arctic sea ice extent in (a) March in historical, (b) September in historical, (c) March in 1pctCO2 and (d) September in 1pctCO2. Blue lines are with snow radiative effects (SoN) and red without (NoS).**

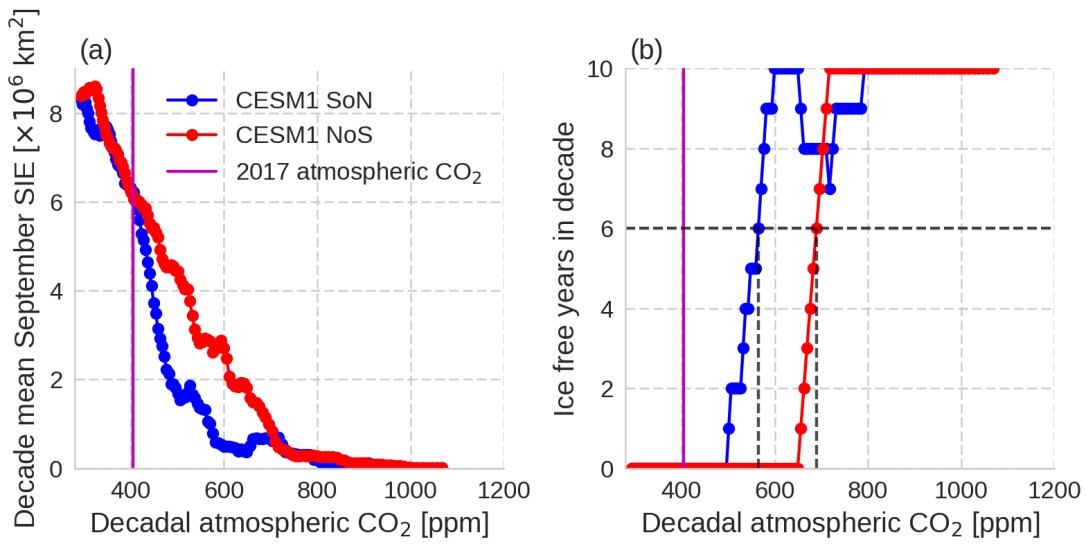

**Figure 3: Changes in September Arctic sea ice under 1 % yr$^{-1}$ CO$_2$ increases for CESM1 SoN (blue) and CESM1 NoS (red) as a function of decade-mean atmospheric CO$_2$. (a) the decadal mean sea ice**

extent and (b) the number of years within that decade for which SIE < 1×10⁶ km², commonly taken as representative of an ice-free Arctic Ocean basin. The atmospheric $CO_2$ concentration in 2017 is shown as a vertical magenta line and the dashed lines in (b) identify the decade-mean atmospheric $CO_2$ level at which the majority of simulated years have SIE < 1×10⁶ km².

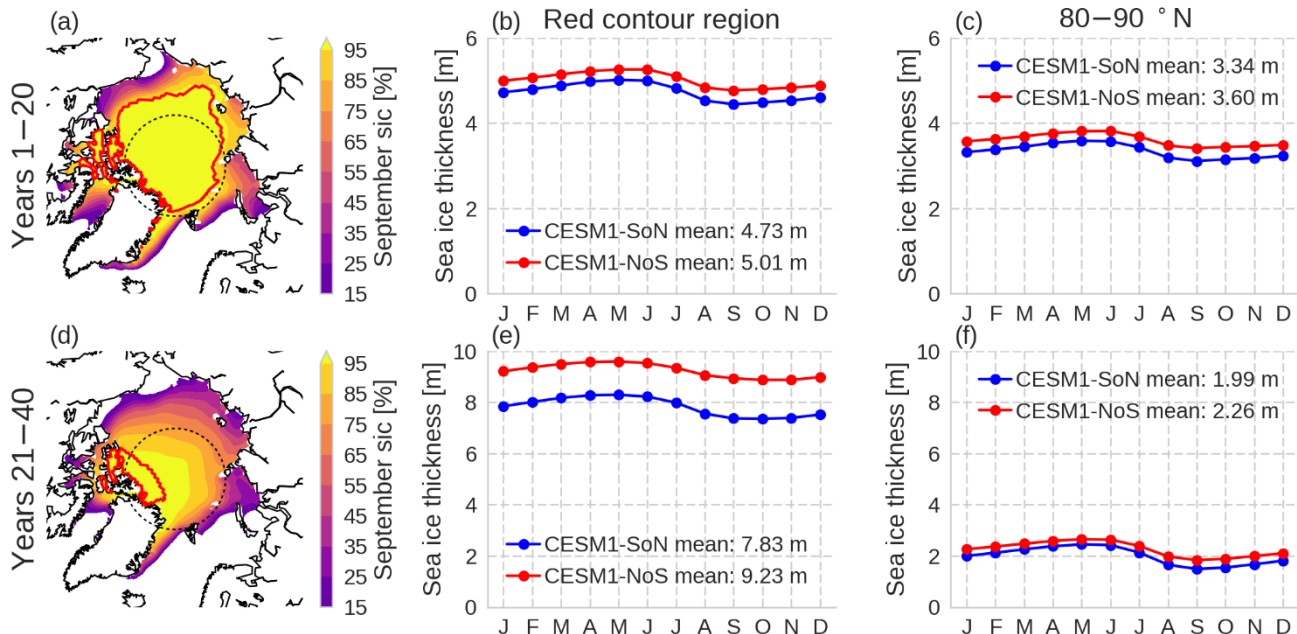

Figure 4: (a) CESM1-SoN September mean sea ice concentration over years 1—20 in 1pct CO2, the black dashed line is 80 °N and the red contour encloses the region within which the mean sea ice concentration exceeds 80 % in all calendar months for both CESM1-SoN and CESM1-NoS. (b) mean thickness within red contour for years 1—20, (c) mean thickness poleward of 80 °N for years 1—20. (d—f) like (a—c) but for years 21—40.

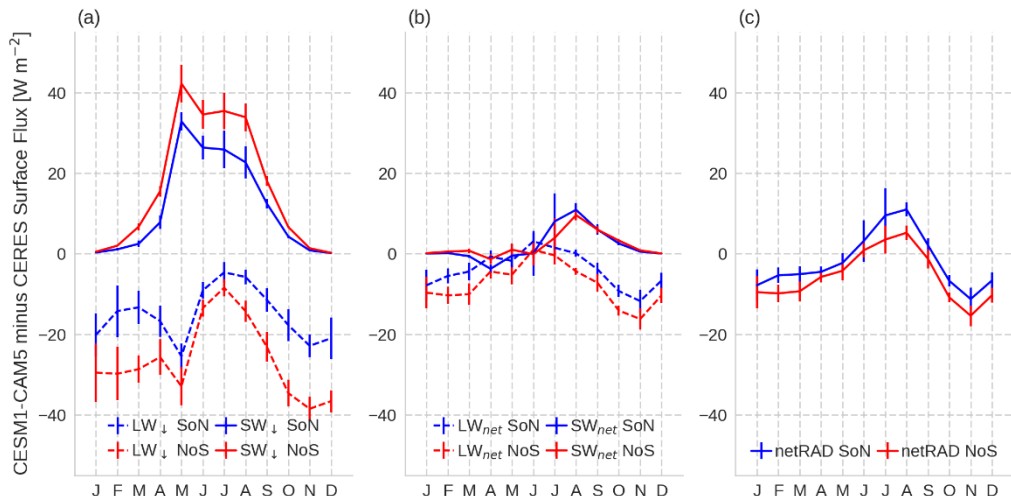

**Figure 5: CESM1 minus CERES 60—90 °N ocean differences in mean surface fluxes for each calendar month over 2001—2005. Differences are shown for both CESM1-SoN (blue) and CESM1-NoS (red). Error bars are estimates of internal variability only, based on standard deviations of non-overlapping 5-year periods in each series after detrending the annual data. (a) Differences in downward longwave (dashed) and downward shortwave (solid). (b) difference in net longwave (dashed, positive downward) and net shortwave (solid). (c) net downward radiation sum. All values are defined such that positive indicates a case where the model value shows greater net downward flux than the CERES value.**

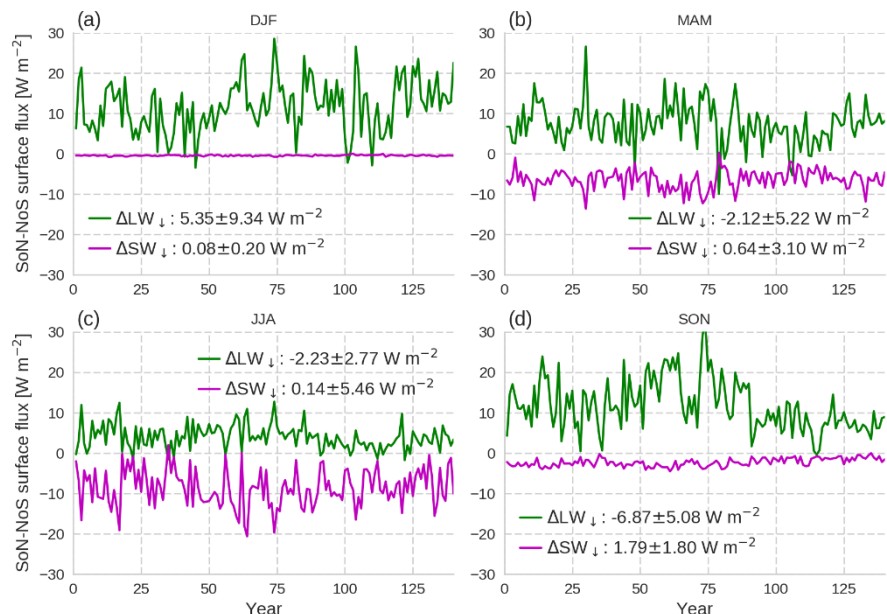

**Figure 6: 1pctCO2 CESM1-SoN minus CESM1-NoS season differences in downward surface fluxes over 60—90 °N oceans. The legend reports the estimate of the 140-year change in this difference by multiplying the linear regression trend coefficient by 140, with ±2σ uncertainties. (a) December-January-February, (b) March-April-May, (c) June-July-August and (d) September-October-November.**

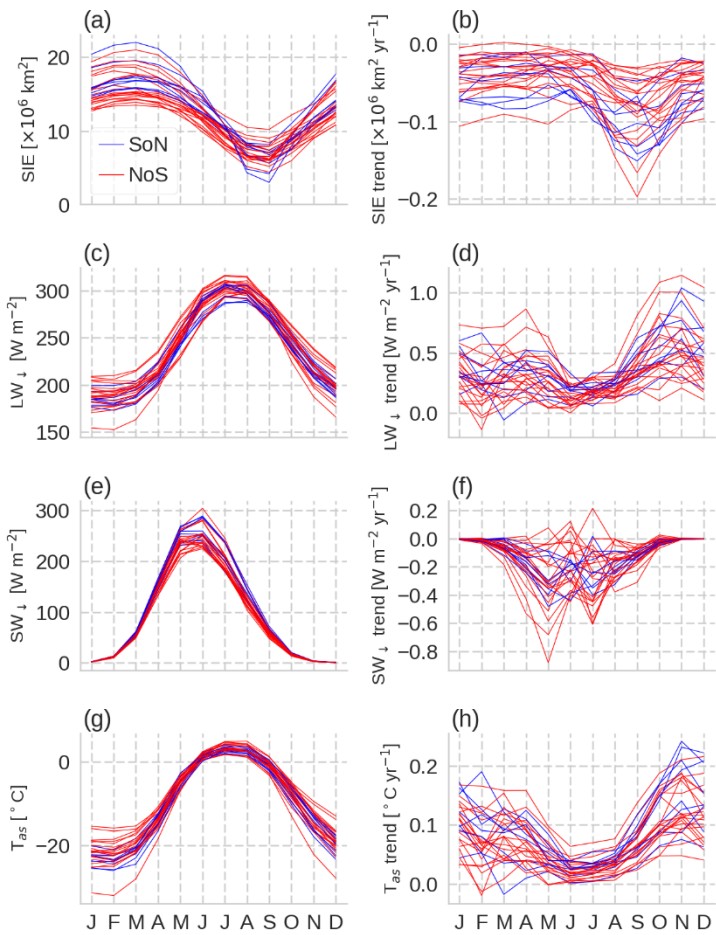

**Figure 7: Output over 60—90 °N oceans from individual CMIP5 historical-RCP8.5 simulations according to whether the simulation includes FIRE (blue) or excludes them (red). Left panels show annual cycles of mean properties from 1979—2005 and right panels show the trend for each calendar month over 2006—2035. (a—b) sea ice extent , (c—d) downward longwave radiation at the surface, (e—f) downward shortwave radiation at the surface and (g—h) near-surface air temperature.**