# Peer review of "Potential faster Arctic sea ice retreat triggered by snowflakes' greenhouse effect"

_The Cryosphere, 2018_

## Referee Comment (RC1) · Anonymous Referee #1 · 16 Nov 2018

General comments:

The paper addresses a relevant topic, which is worth to be published in TC. The overall presentation of the paper is well structured. The language is fluent, but sometimes too colloquial and often not precise enough for my taste. To ensure that the results are reproducible, the methods should be extended. As an example, trends and uncertainties are calculated, but it is often not (or not clearly) written how these are calculated. This makes it difficult to judge whether the statistics are correct. Another aspect that should be improved is testing some of the hypotheses mentioned in the text. I think that this should be easy using the model output of the CESM1-CAM5 simulations (e.g. how the sea ice thickness or how the snow fall changes). Furthermore, more references to the figures would help the reader, it is sometimes not obvious to which figure the text refers

to. Some subfigures are shown, but not discussed.

Specific comments:

- Title: I like the title, but in nearly the whole manuscript, you use the terms "falling ice radiative effects" or "snow radiative effects"; why do you use "snowflakes' greenhouse effect" in the title instead?

- page 1, line 18, "natural factors may have amplified this": Which natural factors, and how can they have amplified the recent Arctic sea ice retreat? Do you mean interannual variability? Instead of "this", I would write "the observed retreat in the last years".

- Page 1, line 23, "(extent $< 1 \times 10^6$ km$^2$)": Please write to what this number refers to. The minimum extent of the year? The extent averaged over some time (September)?

- Page 2, line 23, "Natural atmospheric & ocean dynamics may also contribute...":
  - I would cross the word "natural"
  - I would replace "&" by "and" (in the whole text) if there is no good reason to use "&"
  - please explicitly mention to what the dynamics contribute

- page 2, line 24, "tends to increase extent in winter but ultimately reduce it in summer":
  - "reduce": "reduces"
  - Why does this increase the extent in winter? Because it distributes the sea ice and thus increases the area with a sea ice concentration larger than 15%? Please add at least a reference.

- page 2, line 25, "observations have been used to infer contributions due to anomalously high ice export through...": "observations have been used to infer contributions to summer sea ice reduction from anomalously high ice export through..."?

- page 3, lines 2-3: "the observed extreme low events and general retreating trend have been attributed to a combination of melt driven by global warming along with a likely natural component":

  - Kay et al. (2011) focus on one extreme event, so I would add at least one more reference.
  - I would specify what you mean with natural component. Without context, it could be anything, also a forcing such as volcanic aerosols. You could rephrase the sentence as: "the observed extreme low events and the general retreating trend in summer sea ice extent have been attributed to melt driven by global warming, along with an increased importance of internal variability when sea ice thickness is reduced." (If this is what you mean.)

- page 3, lines 6-9: You directly jump from the attribution to the importance of projections. I would insert the following sentence after "to each forcing.": "A better understanding of the processes that are mainly responsible for sea ice retreat will help to reduce uncertainties in future projections."

- Page 3, lines 13-14, "under high emissions": do you mean here high GHG emissions or a strong forcing? (because anthropogenic aerosol emissions are decreasing in RCP8.5)

- page 3, lines 17-18, "Summer retreat has been faster than the average CMIP5 model simulation, implying a large naturally forced component to recent extremes.":

- I would write "Observed summer retreat".

- I would delete "implying a large naturally forced component to recent extremes" (and the "However" at the beginning of the next sentence). The term "large naturally forced component" is not very meaningful in my opinion. Furthermore, studies imply that internal variability has contributed to the recent extremes, not the fact that the observations show a larger retreat than the models (the models could be wrong due to other issues). In fact, if the models were correct, they would in general be able to simulate that the year-to-year variations in circulation and clouds have a higher impact on sea ice extent when the sea ice thickness is reduced.

• Page 3, line 19: I would replace "forced response" by "sea ice retreat"

• page 3, line 25, "a decrease in surface shortwave which will": "a decrease in downward shortwave radiation, which will" (if this is what you mean)

• page 4, line 1, "a somewhat different expression": "a somewhat different response"?

• Page 4, lines 6-7:

  - "This should manifest later as a faster retreat, both ..." → "This should manifest later as a faster retreat of sea ice area/extent, both...";
  - you could cite here the paper by Massonnet et al. (2018) (https://doi.org/10.1038/s41558-018-0204-z)

• page 4, line 9, "there will be no offset for the stronger expected downward shortwave": "there will be no offset for the weaker expected downward shortwave radiation in summer" (?)

• page 4, line 11, "These effects...":

- – Which effects? Summer versus winter? Reduced downward SW versus lower albedo?
  - – I would cross the "necessarily"
  - – I would write "whether one factor will dominate" instead of "should"

- page 4, line 15, "raise the melting layer": "raise the atmospheric melting layer"

- page 4, line 15-16, "leading to a reduction in the total ice water path (TIWP) in favour of liquid water, which has a smaller radiative effect.": does the "which" refer to "liquid water" or to the "reduction in the total ice water path"?

- page 4 line 23, "We ignore coupled dynamic responses in favour of ...": When I first read this, it sounded to me as if you switched off coupled dynamic responses in your model. After having read the whole paper, I realised that you just wanted to say that you did not analyse potential changes in e.g. ocean heat transport. I would rephrase this sentence.

- page 5, line 4, "for each of the...": this could be misinterpreted, i.e. that you use all ensemble members. I would just cross the "each of"

- page 5, lines 14-15:

  - – "close to 1 degree x 1 degree" → "close to a 1 degree x 1 degree"?
  - – Please say a few more word about these simulations by Li et al. (2014). Do they follow some protocol?

- Page 5, lines 16-17, "and it does this thanks to a two-moment cloud scheme with diagnostic snow":

  - – Our model also has a two-moment cloud scheme and diagnostic snow, but cannot calculate FIRE. I think the important feature of the scheme by Gettelman et al. (2010) is that it treats both the number concentration and the

mixing ratio of snow and rain (instead of only the mass). I would there-
fore rephrase the sentence to: "and it does this thanks to a diagnostic two-
moment treatment of rain and snow"

– Since the whole paper is about FIRE, a few words about how it is calculated
would be beneficial

– "This only represents" → "The scheme only represents"

• page 5, line 19:

– "allows... to be allowed or disallowed": please rephrase

– please mention somewhere in the text explicitly that the only difference be-
tween the simulations CESM2-SoN and CESM1-NoS is switching on/off
FIRE (for both the historical and the 1pctCO2 simulations)

• page 5, lines 21-22:

– "to estimate the first response" → what do you mean here by first response?

– This sounds as if the output were a simulation. You could write: "we use
output of the 1pctCO2 simulation, in which atmospheric $CO_2$ increases at
1% yr -1 for 140 years."

– Please say a bit more about this simulation. Is it a simulation with CMIP5
input/boundary conditions? With what $CO_2$ concentration (corresponding to
which year) does it start? Is this simulation also described in Li et al. (2014)?

• Page 5, line 22, "Radiative forcing definitions differ...": I think you don't mean that
the definitions of the radiative forcing differ (which is also an important question,
e.g. allowing for adjustments or not) but rather that the radiative forcings them-
selves differ?

- Page 5, line 24/25, "We use output for fully coupled CESM1-SoN and for CESM1-NoS runs following the historical and 1pctCO2 simulations.": "We use output from fully coupled CESM1-SoN and for CESM1-NoS runs following the historical and 1pctCO2 scenarios."

- Page 6, line 10-13:

  – Which data did you use for the calculations? The CMIP5 data on the original grid or the data interpolated to a 2.5 degree x 2.5 degree grid?

  – Did you consider the land-sea mask for your calculations (as you did in Section 2.3)? I think that the sea ice concentration from CMIP5 only refers to the oceanic part of the gridbox (at least on the native grids).

- Page 6, lines 19-21:

  – "This combines..." → "CERES-EBAF Surface combines..."

  – "to estimate surface fluxes" → "to calculate surface fluxes"

  – "in each term" → what do you mean with "in each term"? Of each calculated surface flux?

- Page 6, line 22, "previously gridded": "previously interpolated"?

- Page 6, line 23-25:

  – "Fluxes are calculated by taking the area-weighted average of values in each grid cell after scaling by the ocean fraction" → "Fluxes are calculated by taking the area-weighted average after scaling each gridcell by the ocean fraction (including sea ice)"?

  – "we use the CESM1-CAM5 grid" → "we use the CESM1-CAM5 land sea mask"?

- – "a consistent map" → " a consistent fractional land sea mask"?

- Page 6, line 27: "our controlled" → "our historical"?

- page 7, lines 2-5:

  - – I am not sure whether I understand what you did. Did you slice the model output in slices 1979-1982, 1983-1986, 1987-1990, . . . and calculated the standard deviation for each of these slices and then averaged all the standard deviations? And why did you quadrate these values? Maybe a formula or a sketch might be helpful.
  - – The standard deviation of the fluxes might have changed over time, e.g. as a consequence of the sea ice retreat. In my opinion, you could thus just show the standard deviation over the four years of overlap that you have (even if it is large).

- Figures in general: I think it would help the reader if the figures have sublabels (a), (b), etc. that you can refer to.

- Page 7, line 7: "post-1979 changes in SIE": this could be misinterpreted since Figure 1 does not show the changes, but the absolute values in contrast to Supplementary Figs. 3-4

- Page 7, line 11: I would mention the difference between Supplementary Figures 3 and 4.

- page 7, l.12, "The bottom panels of this figure show...generally agrees better with the faster observed retreat": Please mention which figure you mean. I don't see this in Figure 1 (and also not in Supplementary Fig. 2). In March, NoS actually compares better with the observations, and the trend looks similar between NoS and SoN (Fig. 1). In September, NoS is closer to the observations at the beginning, and SoN is closer to them at the end of the observed period. It is hard

to see in Fig. 1 whether the trend in NoS and SoN is different in September. In Supplementary Figs. 3 and 4, it looks like the trend in September is somewhat stronger for SoN. Why don't you calculate the trends for the observations and the CMIP5 medians and compare them? Next to linear regression (which is not very robust), you could also use the Theil-Sen Trend Estimate together with the Mann-Kendall trend test.

- page 7, l.15, "differences in parameterisations for clouds, the atmosphere, oceans...":

  - clouds are a component of the atmosphere, I would not distinguish between the two.
  - Not only parameterisations, but also differences in calculations matter.

- page 7, l. 16:

  - sometimes you write CESM1-CAM5, sometimes only CESM1
  - "controlled" → "historical"?

- Page 7, line 17, "CESM1-CAM5 captures the mean extent well with a smaller discrepancy versus observations throughout the year when including FIRE (full annual cycles in Supplementary Figures 5—6).": You should mention somewhere in the text that the trend in SoN in September is not better than NoS when we compare to the observations since the first is too strong (shown in Supplementary Figure 6). You show in Supplementary Fig. 6 also the observed trend for 1979-2017 so that one could think that the SoN trend in September compares well with the observations. In my opinion, you cannot compare observations by 2017 with simulations by 2005, since it was much warmer between 2005-2017 than before. I would delete this line from the figure (and the text where you mention the trend from 1979 to 2017).

- Page 7, line 18, "full annual cycles...": mention that Supp. Fig. 6 shows trends

- page 7, lines 19-21:

  - how did you calculate the trend and how did you calculate whether the trends differ (you can also write that in the methods)?

  - you could use recursive pre-whitening to account for serial correlation (Wang & Swail 2001, Changes of Extreme Wave Heights in Northern Hemisphere Oceans and Related Atmospheric Circulation Regimes; Zhang & Zwiers 2004, Comment on "Applicability of prewhitening to eliminate the influence of serial correlation on the Mann‐Kendall test")

  - I think it is sufficient to provide the p-value, t gives no real information (?)

- page 7, line 21:

  - "Neither show significant differences relative..." → "Neither are differences significant relative..."?

- page 7, line 23, "the bottom panels show": of which figure?

- page 8, line 3, "majority of years...": It would be helpful to add a dashed line in Fig.3b at the year when the majority of years (i.e. 6 years) are ice-free (and down to the corresponding $CO_2$ values)

- page 8, line 4, "In an naïve sense this implies...": I thought that the relationship between cumulative $CO_2$ emissions and the $CO_2$ concentration in the atmosphere is not linear. Or is the approximation of a linear function valid for the time scales that you are looking at?

- page 8, line 8, "a more rapid collapse of Arctic sea ice in reality": more rapid than what? Than previously simulated by CMIP5 models?

[Figure]

- page 9, line 1, "Absorbed longwave dominates": absorbed the by surface? And dominates over what? Absorbed shortwave radiation (where is this shown)?

- Page 9, lines 1-2, "CESM1-SoN's lower SIE results in a lower albedo that more than offsets the reduced SW downward such that absorbed SW is also higher when including FIRE.":

  – "CESM1-SoN's lower SIE results in a lower albedo that more than offsets the reduced SW downward such that SW absorbed at the surface is also higher when including FIRE."

  – This explains why the difference in SW between SoN and NoS in Fig. 4b is not large, correct? If yes, I would explictly refer to this subfigure.

- Page 9, line 3-5:

  – "on average": yearly average?

  – I think that changes in the net radiation matter more than the downward longwave radiation (?).

  – Please also discuss Fig. 4c. It shows that the difference in the net downward radiation sum between the model and the observation is smaller for many months, but larger in September with SoN. Please also think about how to use the word "net"; for Fig. 4b, you use "net" as downward+upward; for Fig. 4c, you use "net" as LW+SW.

  – Figure 4c shows the sum of LW and SW shown in 4a if I understand the caption correctly. However, if I simply add the values in a, I don't get the same values as in 4c. Did I misinterpret the figure?

- Page 9, line 7, "This would manifest as...":

  – please mention here that you now switch to the 1pctCO2 simulations

- – "differences in time" → "differences over time"?

- page 9, lines 10-11:

  - – "changes are estimated" → "trends are estimated" (to be more precise because you sometimes also look at changes between two simulations or changes between observations and simulations)
  - – please mention how you calculated the trend
  - – "changes occur" → "trend occurs"

- page 9, line 13: what is the plus/minus refering to?

- page 9, line 14, "so this change": "so the following change in trend"

- page 9, line 15, "by year 70": refer to the figure

- page 9, line 18: why do you use a range of 14-86% here? in other occasions you showed 10-90% percentiles or 2*sigma

- page 9, line 19-21:

  - – Could the following maybe also be an explanation: when there is sea ice in NoS, but no sea ice left in SoN, I expect that the cloud radiative effect in SoN is larger because there is more evaporation from the ocean's surface. When later both NoS and SoN are ice-free, the cloud radiative effect (and the downward LW radiations) would be more similar.
  - – Can you diagnose the transition from snow to rain from the model output to confirm your hypothesis?
  - – Are the radiative properties of rain also considered in your model or are these totally negligible?

- page 9, line 25: does your simulated output confirm that the sea ice thickness becomes thinner?

- Page 10, lines 13-14, "two models that include FIRE show substantially more summertime SW...":

  – more than what (CMIP5 median)?
  – Can you show this somewhere or provide some numbers?

- page 11, line 1, "too much surface shortwave radiation": "too much downward shortwave radiation"?

- page 11, lines 17-20: Can you calculate from your model output how much sea ice has melted in your simulations (in SoN and NoS)?

- Page 11, lines 21-23: Why did you actually not look at least at some other variables? As an example, it should be easy to see how different the clouds and precipitation are between the two simulations (e.g. liquid water path, cloud cover, snow versus rain).

- page 11, line 26, "lead to counteracting processes": do you mean: "may disperse the snow radiative effect"?

- Page 12, line 1, being approximately twice as fast: Do you show that somewhere in the paper? How many years from now on for the two cases?

- Figures in general: Sometimes you use parentheses and sometimes square brackets around the units.

- Figure 1, caption, 10-90% range: I would write "10-90% percentile range" (in the whole text) to be more precise

- Figure 2, caption: "and" before CESM1-CAM5

- Figure 3, caption: please delete "but any comparison must be carefully made ...". In my opinion, statements like this do not belong to a caption but only to the main text.

- Figure 5:

- caption: mention that this figure shows 1pctCO2

- The units should be W/m$^2$.

- Figure 6, caption: delete "poleward of 30 degree" since you show output between 60 and 90 degree N

- Supplementary Material, Table 1:

- "whether they exclude falling ice radiative effects": this sounds as if the models have FIRE implemented but exclude them; how about "neglect falling ice radiative effects"?

- "this subset is all those for whom": please rewrite, e.g. "All r1i1p1 simulations were considered that provide the scenarios of interest and the necessary output of surface fluxes and sea ice fields."

- Supplementary Figure 1:

  – to what do the colour of the points correspond to (seasons)?
  – If there are more than 8 simulations that you compared, you could add in the caption that the other plots look similar (if this is the case)

- Supplementary Figure 3, caption: first you write that the anomaly is relative to 1979-1984, then you write that you calculated the anomalies relative to 1979 (?)

- Supplementary Figure 4, "SIE change is shown as a fraction relative to its 1979-1984 mean": I would rather write that Supplementary Figure 4 shows relative changes (instead of absolute changes).

- Supplementary Figure 5:

- "No uncertainties are shown...": You could detrend the time series before you calculate the standard deviation.

- Supplementary Figure 6, "and may be an underestimate...":

  – I would not write that in the caption but discuss it in the text.
  – Do the lag-1 correlations that you mention refer to individual months? If yes, could you calculate the trend considering the lag-1 correlation for each month individually? Does it make a large difference if you account for autocorrelation? How does it change if you take another trend estimator than linear regression?
  – Please mention how you calculate the sigma. Is this the standard deviation of the white noise? Or is it the uncertainty of the trend (which would be more important from my point of view)?
  – The error bars overlap for many months and therefore it is impossible to see the standard deviations.

Technical corrections:

- Page 1, line 24, "downward shortwave": I would (always in the paper) write "downward shortwave radiation" (the same of course for longwave)

- Page 2, lines 10-11, "Physically, ice affects both...": Physically, sea ice affects both..."

- Page 2, line 13, "From a surface perspective": the previous sentence also refers to the surface

- Page 2, line 14, "sea-ice extent": "sea ice extent"

- page 3, line 1, "From analyses of subsets of climate models in the Climate Model Intercomparison Project, phase 5 (CMIP5 (Taylor et al., 2012)), ...": This sentence sound complicated. Why not: "Based on CMIP5 data (Climate Model Intercomparison Project, phase 5; Taylor et al., 2012), the observed ..."

- page 3, line 7, "are also necessary": "are necessary"

- page 3, line 22, "tends": "tend"

- Page 4, line 5, "increased winter longwave": "increased winter longwave downward radiation"

- page 4, line 9-10, "This will mean that a non-FIRE simulation should experience more local albedo feedback due to...": "This will mean that a non-FIRE simulation should experience a stronger local snow-albedo feedback due to..."

- page 4, line 16-17, "the direct effect": "the direct consequence"? (because of "radiative effect" in the previous sentence)

- page 5, line 3, "who have": "that provide"

- Page 5, line 7, "This is a scenario of very high radiative forcing which we select...": comma before "which"

- page 5, line 11 (and in general): you use FIRE as a singular but is it not a plural ("falling ice radiative effects")?

- Page 5, line 12, "and those in which there are no snow radiative effects": "and those in which snow radiative effects are not considered"

- page 5, line 13, "These are listed...": "All models are listed..."

- Page 7, line 19: delete "also"

- Page 8, line 1, "decadal mean SIE": "decadal mean September SIE"

- page 8, line 7, "potential magnitude": "potential impact"?

- page 8, line 14, "in future models": "in future model versions"?

- page 9, line 17: "healthy" sounds colloquial to me

- Page 11, line 12, "shows": show

---

## Referee Comment (RC2) · Devasthale (Referee) · 30 Nov 2018

Review of Li et al:

I will keep it short and to the point. Li et al bring in an important aspect into discussion here, i.e. the radiative effects (particularly longwave warming) of falling snow. From the process point of view, I do appreciate that the authors highlight its potential importance and encourage modelling community to take this process into account. The manuscript is written and presented nicely. The analysis is robust and the arguments are justified well based on the results presented here. I do however have few major comments.

1) The overwhelming focus on the radiative effects, by neglecting the dynamical and surface aspects, concerns me. I understand that the authors neglect them for the sake of simplicity, but they are actually important here. For example, between the two sets of CMIP5 models, SoN and NoS, the former shows more realistic trends in sea-ice extent. Could it be a coincidence? How much of it is really down to including FIRE and/or down to having different dynamical responses and surface descriptions in these sets of models? Please note that CMIP5 models vary widely in their description of sea-ice (e.g. Koenigk et al., 2014). Could the authors please check how the SoN and NoS models differ in these aspects?

2) FIRE would depend not only on how much it precipitates, but also on the frequency of falling snow. But there seems to be hardly any discussion about this (and how it varies across NoS and SoN). Or am I missing something here?

I hope the authors comment on these issues.

References

Koenigk, T., Devasthale, A., and Karlsson, K.-G.: Summer Arctic sea ice albedo in CMIP5 models, Atmos. Chem. Phys., 14, 1987-1998, https://doi.org/10.5194/acp-14-1987-2014, 2014.

---

## Author Comment (AC1) · 16 Jan 2019

Dear Editor and two Reviewers,

Please find the attached zip file ( in tc-2018-195-supplement) which includes the red-lined/clean version of the main text and SI files as well as the point-by-point responses for two Reviewers (RC1 & RC2).

We thank both reviewers for their insightful comments and clear, detailed analysis of our paper. We have attached a redlined draft that shows the changes along with a point-by-point response to each reviewer.

On behalf of all authors,

[Figure]

Jui-Lin F. Li

Please also note the supplement to this comment:
https://www.the-cryosphere-discuss.net/tc-2018-195/tc-2018-195-AC1-supplement.zip

———————————————

---

## Author Comment (AC2) · 16 Jan 2019

Dear Reviewer No.1,

Please find the attached zip file (tc-2018-195-RESPONSE-All) which includes the red-lined/clean version of the main text and SI files as well as the point-by-point responses for two Reviewers (RC1 & RC2).

We thank both reviewers for their insightful comments and clear, detailed analysis of our paper. We have attached a redlined draft that shows the changes along with a point-by-point response to two Reviewers' comments.

On behalf of all authors,

[Figure]

Jui-Lin F. Li

Please also note the supplement to this comment:
https://www.the-cryosphere-discuss.net/tc-2018-195/tc-2018-195-AC2-supplement.zip
———————————————————————————

---

## Author Response (AR1)

Anonymous Referee #1 Received and published: 16 November 2018 General comments:

- 10 The paper addresses a relevant topic, which is worth to be published in TC. The overall presentation of the paper is well structured. The language is fluent, but sometimes too colloquial and often not precise enough for my taste. To ensure that the results are re- producible, the methods should be extended. As an example, trends and uncertainties are calculated, but it is often not (or not clearly) written how these are calculated. This makes it difficult to judge whether the statistics are correct. Another aspect that should be improved is testing some
- 15 of the hypotheses mentioned in the text. I think that this should be easy using the model output of the CESM1-CAM5 simulations (e.g. how the sea ice thickness or how the snow fall changes). Furthermore, more references to the figures would help the reader, it is sometimes not obvious to which figure the text refers to. Some subfigures are shown, but not discussed.
- 20 We thank the reviewer for a thorough and thoughtful analysis of our submission. The paper is now longer, but we think that the main points are both clearer and better supported. We are grateful for advice that led to an improved the manuscript.

We now state in the intro that our main points are (1) we test whether FIRE affect simulated sea ice retreat enough to be worth highlighting to model developers, and (2) is there evidence in support of our idea that

25 FIRE thin the initial pack and means faster future retreat. We use this to justify our focus on the radiative terms.

Specific comments:

• Title: I like the title, but in nearly the whole manuscript, you use the terms "falling ice radiative effects" or "snow radiative effects"; why do you use "snowflakes' greenhouse effect" in the title instead? **Comments:** We wish to emphasise the longwave component while keeping a shorter title. "Snowflakes' greenhouse effect" is snappier than "falling snow longwave radiative effects"

5 Changes: N/A

• page 1, line 18, "natural factors may have amplified this": Which natural factors, and how can they have amplified the recent Arctic sea ice retreat? Do you mean interannual variability? Instead of "this", I would write "the observed retreat in the last years".

10 **Comments:**

Changes: Change made.

• Page 1, line 23, "(extent  $< 1 \times 10^6 \text{ km}^2$ )": Please write to what this number refers to. The minimum extent of the year? The extent averaged over some time (September)?

15 Comments: We intended that month may be "ice free" but expect September to be the first occurrence. Changes: We have changed to "monthly mean extent".

• Page 2, line 23, "Natural atmospheric & ocean dynamics may also contribute...":

-I would cross the word "natural"

-I would replace "&" by "and" (in the whole text) if there is no good reason to use "&"
 -please explicitly mention to what the dynamics contribute

**Comments:**

**Changes:** New text: "Atmosphere and ocean dynamics may also export ice to lower latitudes. For example, stronger circulation associated with the Arctic Oscillation can increase the total area of new, thin ice but

25 transport the thicker ice away from the coldest regions and leave it vulnerable to summer melting (Rigor et al., 2002)".

"&" replaced throughout.

• page 2, line 24, "tends to increase extent in winter but ultimately reduce it in summer":

-"reduce": "reduces"

-Why does this increase the extent in winter? Because it distributes the sea ice and thus increases the area with a sea ice concentration larger than 15%? Please add at least a reference.

**5 **Comments:**

**Changes: See change above.**

• page 2, line 25, "observations have been used to infer contributions due to anomalously high ice export through...": "observations have been used to infer contributions to summer sea ice reduction from anomalously

10 high ice export through..."?

**Comments:**

Changes: Change made.

• page 3, lines 2-3: "the observed extreme low events and general retreating trend have been attributed to a

15 combination of melt driven by global warming along with a likely natural component":

-Kay et al. (2011) focus on one extreme event, so I would add at least one more reference.

-I would specify what you mean with natural component. Without context, it could be anything, also a forcing such as volcanic aerosols. You could rephrase the sentence as: "the observed extreme low events and the general retreating trend in summer sea ice extent have been attributed to melt driven by global warming, along

20 with an increased importance of internal variability when sea ice thickness is reduced." (If this is what you mean.)

**Comments:** The original phrasing was meant to allow the possibility that changes in e.g. clouds and circulation could be natural, or could also be a coupled response to forcing.

Changes: we have rephrased and use Kay et al. as an example of cloud anomalies and Rigor & Wallace 2004

as an example of how circulation may have primed the pack for loss.

• page 3, lines 6-9: You directly jump from the attribution to the importance of projections. I would insert the following sentence after "to each forcing.": "A better understanding of the processes that are mainly responsible for sea ice retreat will help to reduce uncertainties in future projections."

**Comments:**

10

5 Changes: Change made, without "mainly".

• Page 3, lines 13-14, "under high emissions": do you mean here high GHG emissions or a strong forcing? (because anthropogenic aerosol emissions are decreasing in RCP8.5)

**Comments:** Good point, but we wish to allow for cases of low GHG emissions but with strong carbon cycle feedbacks too.

Changes: Term changed to "radiative forcing".

• page 3, lines 17-18, "Summer retreat has been faster than the average CMIP5 model simulation, implying a large naturally forced component to recent extremes.":-I would write "Observed summer retreat".-I would

- 15 delete "implying a large naturally forced component to recent extremes" (and the "However" at the beginning of the next sentence). The term "large naturally forced component" is not very meaningful in my opinion. Furthermore, studies imply that internal variability has contributed to the recent extremes, not the fact that the observations show a larger retreat than the models (the models could be wrong due to other issues). In fact, if the models were correct, they would in general be able to simulate that the year-to-year variations in circulation
- 20 and clouds have a higher impact on sea ice extent when the sea ice thickness is reduced.

**Comments:**

Changes: We removed the suggested text and link the two sentences with "and".

- Page 3, line 19: I would replace "forced response" by "sea ice retreat"
- 25 **Comments:**

Changes: Change made.

• page 3, line 25, "a decrease in surface shortwave which will": "a decrease in downward shortwave radiation, which will" (if this is what you mean)

**Comments:**

**Changes:** Change made and now term "SW↓" is introduced here.

**5**

• page 4, line 1, "a somewhat different expression": "a somewhat different response"?

**Comments:**

Changes: We feel either could work. Change made.

Page 4, lines 6-7:-"This should manifest later as a faster retreat, both ..."→"This should manifest later as a faster retreat of sea ice area/extent, both...";

-you could cite here the paper by Massonnet et al. (2018) (https://doi.org/10.1038/s41558-018-0204-z)

Comments: This is a nice paper, it fits neatly with our argument so we use it as a reference throughout.

Changes: Citation added and our discussion extended to discuss its findings.

**15**

• page 4, line 9, "there will be no offset for the stronger expected downward shortwave": "there will be no offset for the weaker expected downward shortwave radiation in summer" (?)

**Comments:**

Changes: This has been rephrased to focus on the local SW albedo feedback only.

**20**

• page 4, line 11, "These effects...":-Which effects? Summer versus winter? Reduced downward SW versus lower albedo?

-I would cross the "necessarily"

-I would write "whether one factor will dominate" instead of "should"

**25 **Comments:**

**Changes:** Changed to "The SW $\downarrow$  and LW $\downarrow$  effects from including FIRE should oppose each other and it is not necessarily obvious whether one factor will dominate."

• page 4, line 15, "raise the melting layer": "raise the atmospheric melting layer"

**Comments:**

Changes: Change made.

5 • page 4, line 15-16, "leading to a reduction in the total ice water path (TIWP) in favour of liquid water, which has a smaller radiative effect.": does the "which" refer to "liquid water" or to the "reduction in the total ice water path"?

**Comments:** This was originally to refer to a switch from falling snow to falling rain which would remove modelled TIWP and place it into the rain component, which does not interact with radiation.

10 Changes: We have rephrased.

• page 4 line 23, "We ignore coupled dynamic responses in favour of ...": When I first read this, it sounded to me as if you switched off coupled dynamic responses in your model. After having read the whole paper, I realised that you just wanted to say that you did not analyse potential changes in e.g. ocean heat transport. I

15 would rephrase this sentence.

**Comments:**

**Changes: This has been rephrased.**

• page 5, line 4, "for each of the...": this could be misinterpreted, i.e. that you use all ensemble members. I 20 would just cross the "each of"

**Comments:**

Changes: Done.

- page 5, lines 14-15:
- 25 -"close to 1 degree x 1 degree"  $\rightarrow$  "close to a 1 degree x 1 degree"?

-Please say a few more word about these simulations by Li et al. (2014). Do they follow some protocol?

**Comments:**

Changes: We now state that these follow CMIP5 protocol for both historical and 1pctCO2.

• Page 5, lines 16-17, "and it does this thanks to a two-moment cloud scheme with diagnostic snow":

-Our model also has a two-moment cloud scheme and diagnostic snow, but cannot calculate FIRE. I think the important feature of the scheme by Gettelman et al. (2010) is that it treats both the number concentration and

5 the mixing ratio of snow and rain (instead of only the mass). I would therefore rephrase the sentence to: "and it does this thanks to a diagnostic two-moment treatment of rain and snow"

-Since the whole paper is about FIRE, a few words about how it is calculated would be beneficial

-"This only represents"→"The scheme only represents"

**Comments:**

10 Changes: New text including:

"Falling snow mass and the crystal number concentration is diagnosed at each model level and time step, and is related to an effective radius as detailed in Section 2 of Morrison and Gettelman (2008). The profile of snow mass and effective radius is then related to radiative properties using precomputed lookup tables based on an assumed ice habit mixture as described in Section 2.5 of Gettelman et al. (2010)."

**15**

• page 5, line 19:

-"allows... to be allowed or disallowed": please rephrase

-please mention somewhere in the text explicitly that the only difference between the simulations CESM2-SoN and CESM1-NoS is switching on/off FIRE (for both the historical and the 1pctCO2 simulations)

**20 **Comments:**

Changes: Change made and clarification added to the end of this sentence.

• page 5, lines 21-22:

-"to estimate the first response"→what do you mean here by first response?

25 -This sounds as if the output were a simulation. You could write: "we use output of the 1pctCO2 simulation, in which atmospheric CO2 increases at 1% yr -1 for 140 years." -Please say a bit more about this simulation. Is it a simulation with CMIP5 input/boundary conditions? With what CO2 concentration (corresponding to which year) does it start? Is this simulation also described in Li et al. (2014)?

Comments: Oops, we meant "forced". The implementation is described in Li et al. (2014) but 1pctCO2 is not

5 used there. We think our previous change that discusses CMIP5 protocol is enough for readers to understand.Changes: "first" replaced with "forced".

• Page 5, line 22, "Radiative forcing definitions differ...": I think you don't mean that the definitions of the radiative forcing differ (which is also an important question, e.g. allowing for adjustments or not) but rather

10 that the radiative forcings themselves differ?

**Comments:** We wished to express that radiative forcings definitions differ (e.g. depending on which adjustments are included) and also that calculations for doubled CO2 differ (e.g. fixed SST versus Gregory plot, and if you use Gregory approach then over what period do you regress?).

Changes: We replaced "definitions" with "estimates".

15

• Page 5, line 24/25, "We use output for fully coupled CESM1-SoN and for CESM1-NoS runs following the historical and 1pctCO2 simulations.": "We use output from fully coupled CESM1-SoN and for CESM1-NoS runs following the historical and 1pctCO2 scenarios."

**Comments:**

20 Changes: Change made.

• Page 6, line 10-13:

-Which data did you use for the calculations? The CMIP5 data on the original grid or the data interpolated to a 2.5 degree x 2.5 degree grid?

25 – Did you consider the land-sea mask for your calculations (as you did in Section 2.3)? I think that the sea ice concentration from CMIP5 only refers to the oceanic part of the gridbox (at least on the native grids).
 Comments: We used OImon/sic and fx/areacello, thus accounting for the ocean covered area only.
 Changes: New text: "total area of all of the model's native ocean grid cells with sic > 15 %"

• Page 6, lines 19-21:

-"This combines..."→"CERES-EBAF Surface combines..."

-"to estimate surface fluxes"→"to calculate surface fluxes"

5 –"in each term"→what do you mean with "in each term"? Of each calculated surface flux?

**Comments:**

Changes: Changes made, including to "in each surface radiative flux term".

• Page 6, line 22, "previously gridded": "previously interpolated"?

**10 **Comments:**

Changes: Done.

• Page 6, line 23-25:

-"Fluxes are calculated by taking the area-weighted average of values in each grid cell after scaling by the

15 ocean fraction"→"Fluxes are calculated by taking the area-weighted average after scaling each gridcell by the ocean fraction (including sea ice)"?

-"we use the CESM1-CAM5 grid"→"we use the CESM1-CAM5 land sea mask"?

-"a consistent map"  $\rightarrow$  " a consistent fractional land sea mask"?

**Comments:**

20 Changes: Done.

• Page 6, line 27: "our controlled"→"our historical"?

**Comments:** We use "controlled" to mean our CESM simulations in which we control whether FIRE are allowed.

25 **Changes:** Introduction text added: "We refer to these as our "controlled" simulations to emphasise that we controlled the inclusion of FIRE and to distinguish them from other studies' CESM1 simulations."

• page 7, lines 2-5:

-I am not sure whether I understand what you did. Did you slice the model output in slices 1979-1982, 1983-1986, 1987-1990,...and calculated the standard deviation for each of these slices and then averaged all the standard deviations? And why did you quadrate these values? Maybe a formula or a sketch might be helpful. -The standard deviation of the fluxes might have changed over time, e.g. as a consequence of the sea ice

5 retreat. In my opinion, you could thus just show the standard deviation over the four years of overlap that you have (even if it is large).

**Comments:** Since we are comparing 5-year means (we put 4-, this was a typo and has been corrected), we wish to have the difference between 5-year means and the standard deviation of the sampling distribution of the 5-year mean. We estimate it by taking non-overlapping 5-year periods and then taking their standard

10 deviation. Quadrature is needed because we are looking at the model minus obs difference so need to combine their uncertainties.

Changes: Typo corrected for 4-year versus 5-year.

A new paragraph in Section 2.3 explains our approach.

• Figures in general: I think it would help the reader if the figures have sublabels (a), (b), etc. that you can refer to.

**Comments:**

**Changes: These have been added.**

Page 7, line 7: "post-1979 changes in SIE": this could be misinterpreted since Figure 1 does not show the changes, but the absolute values in contrast to Supplementary Figs. 3-4
 Comments:

Changes: We now just say "post-1979 SIE".

Page 7, line 11: I would mention the difference between Supplementary Figures 3 and 4.

**Comments:**

Changes: Text in parentheses rewritten, and these are now supplementary figures 4-5.

• page 7, 1.12, "The bottom panels of this figure show...generally agrees better with the faster observed retreat": Please mention which figure you mean. I don't see this in Figure 1 (and also not in Supplementary Fig. 2). In March, NoS actually compares better with the observations, and the trend looks similar between NoS and SoN

- 5 (Fig. 1). In September, NoS is closer to the observations at the beginning, and SoN is closer to them at the end of the observed period. It is hard to see in Fig. 1 whether the trend in NoS and SoN is different in September. In Supplementary Figs. 3 and 4, it looks like the trend in September is somewhat stronger for SoN. Why don't you calculate the trends for the observations and the CMIP5 medians and compare them? Next to linear regression (which is not very robust), you could also use the Theil-Sen Trend Estimate together with the Mann-
- 10 Kendall trend test.

Comments: We like this suggestion a lot, we find that our results are robust when using Theil-Sen.

**Changes:** New Supplementary Figure 6 shows OLS (stationary Gaussian white noise assumed) and Theil-Sen fits for 1979—2017.

Figure 1 discussion added: "Trend analysis shows that the median CMIP5-SoN retreat is visibly greater than 15 CMIP5-NoS from June through October, in better agreement with observations (Supplementary Figure 6)."

• page 7, 1.15, "differences in parameterisations for clouds, the atmosphere, oceans...":

- clouds are a component of the atmosphere, I would not distinguish between the two.

-Not only parameterisations, but also differences in calculations matter.

20 **Comments:**

**Changes:** Changed to "differences in parameterisations and calculation methods for the atmosphere, oceans and sea ice..."

• page 7, l. 16:

25 –sometimes you write CESM1-CAM5, sometimes only CESM1 –"controlled"→"historical"?

**Comments:**

Changes: See previous for "controlled"

All uses of CESM1 are now CESM1-CAM5, CESM1-SoN or CESM1-NoS, with the implication being that the SoN and NoS cases use CAM5. Except when referring to CMIP5 or the large ensemble.

• Page 7, line 17, "CESM1-CAM5 captures the mean extent well with a smaller discrepancy versus

- 5 observations throughout the year when including FIRE (full annual cycles in Supplementary Figures 5â: You should mention somewhere in the text that the trend in SoN in September is not better than NoS when we compare to the observations since the first is too strong (shown in Supplementary Figure 6). You show in Supplementary Fig. 6 also the observed trend for 1979-2017 so that one could think that the SoN trend in September compareswell with the observations. In my opinion, you cannot compare observations by 2017
- 10 with simulations by 2005, since it was much warmer between 2005-2017 than before. I would delete this line from the figure (and the text where you mention the trend from 1979 to 2017)

**Comments:**

**Changes:** Supplementary figure line deleted. Main text now mentions the 1979—2005 SoN-obs trend p = 0.06 and says that while observed loss rates increased after 2005, we can't do a direct comparison with the

15 available output.

• Page 7, line 18, "full annual cycles...": mention that Supp. Fig. 6 shows trends

**Comments:**

Changes: This has been rewritten with separate sentences for mean extent and trends.

**20**

• page 7, lines 19-21:

-how did you calculate the trend and how did you calculate whether the trends differ (you can also write that in the methods)?

-you could use recursive pre-whitening to account for serial correlation (Wang & Swail 2001, Changes of

25 Extreme Wave Heights in Northern Hemisphere Oceans and Related Atmospheric Circulation Regimes; Zhang & Zwiers 2004, Comment on "Applicability of prewhitening to eliminate the influence of serial correlation on the Mann-Kendall test")

-I think it is sufficient to provide the p-value, t gives no real information (?)

**Comments:** We took a more concise approach. We use Ljung-Box for detection of autocorrelation and see no strong evidence for non-white noise. The lag-1 autocorrelations that are significant in one period are not in the other, and are actually negative which means we are being cautious by using white-noise-based errors. Also, the results are similar for Theil-Sen confidence bounds.

5 **Changes:** New paragraph in Section 2.2 and Supplementary Table 2 with statistics calculated from the OLS trend residuals applied to NSIDC SIE.

Later mentions of autocorrelation deleted, we mention that we assume white noise and that Theil-Sen estimates are similar.

t-value deleted.

- 10
- page 7, line 21:

-"Neither show significant differences relative..."→"Neither are differences significant relative..."?

**Comments:**

Changes: paragraph has been largely rewritten.

15

• page 7, line 23, "the bottom panels show": of which figure?

**Comments:**

**Changes:** We now refer to panels, in this case Figure 2(d).

- 20 page 8, line 3, "majority of years...": It would be helpful to add a dashed line in Fig.3b at the year when the majority of years (i.e. 6 years) are ice-free (and down to the corresponding CO2 values)
   Comments:
   Changes: Lines added.
- page 8, line 4, "In an naïve sense this implies...": I thought that the relationship between cumulative CO2 emissions and the CO2 concentration in the atmosphere is not linear. Or is the approximation of a linear function valid for the time scales that you are looking at?

**Comments:** We intended "naïve" to imply the conclusion following roughly linear assumptions because carbon cycle feedbacks are a massive potential maze.

**Changes:** We have added a citation to Matthews et al. whose Figure 2(a) shows pretty constant airborne fraction under 1pctCO2 for years 50—70.

**5**

• page 8, line 8, "a more rapid collapse of Arctic sea ice in reality": more rapid than what? Than previously simulated by CMIP5 models?

**Comments:**

Changes: Indeed, fixed.

**10**

20

• page 9, line 1, "Absorbed longwave dominates": absorbed the by surface? And dominates over what? Absorbed shortwave radiation (where is this shown)?

**Comments:**

Changes: Paragraph now discusses each panel of Figure 4, e.g.: "From Figure 4(b), the net absorbed surface

15 SW radiation shows relatively small SoN-NoS differences because while FIRE reduces SW↓, it also reduces SIE and so lowers the mean albedo. The net absorbed surface longwave radiation is consistently greater in SoN, explaining the majority of the remaining difference in net radiation in Figure 4(c)."

• Page 9, lines 1-2, "CESM1-SoN's lower SIE results in a lower albedo that more than offsets the reduced SW downward such that absorbed SW is also higher when including FIRE.":

-"CESM1-SoN's lower SIE results in a lower albedo that more than offsets the reduced SW downward such that SW absorbed at the surface is also higher when including FIRE."

-This explains why the difference in SW between SoN and NoS in Fig. 4b is not large, correct? If yes, I would explicitly refer to this subfigure.

**25 **Comments:**

Changes: See above.

• Page 9, line 3-5:

-"on average": yearly average?

-I think that changes in the net radiation matter more than the downward longwave radiation (?).

-Please also discuss Fig. 4c. It shows that the difference in the net downward radiation sum between the model and the observation is smaller for many months, but larger in September with SoN. Please also think about

5 how to use the word "net"; for Fig. 4b, you use "net" as downward+upward; for Fig. 4c, you use "net" as LW+SW.

-Figure 4c shows the sum of LW and SW shown in 4a if I understand the caption correctly. However, if I simply add the values in a, I don't get the same values as in 4c. Did I misinterpret the figure?

**Comments:** We did not explain this clearly enough! We use "net" consistently to be down minus up, i.e. netRAD" is net SW + net LW, so you can't use Figure 4(a) to make Figure 4(c), you must use Figure 4(b).

**Changes:** The caption changed, now ends with "All values are defined such that positive indicates that the model shows greater net downward flux than CERES."

Main text changed to better describe this.

15 • Page 9, line 7, "This would manifest as...":

please mention here that you now switch to the 1pctCO2 simulations

-"differences in time"→"differences over time"?

**Comments:**

**Changes: Done.**

**20**

10

-"changes are estimated"  $\rightarrow$  "trends are estimated" (to be more precise because you sometimes also look at changes between two simulations or changes between observations and simulations)

-please mention how you calculated the trend

25 -"changes occur"  $\rightarrow$  "trend occurs"

**Comments:**

**Changes:** Text changes made, and we added "OLS" to the sentence "multiplying the OLS trend gradient". This acronym for optimised least squares is introduced in our new methods Section 2.2.

• page 9, lines 10-11:

• page 9, line 13: what is the plus/minus refering to?

**Comments:**

Changes: Text added to clarify.

**5**

• page 9, line 14, "so this change": "so the following change in trend"

**Comments:**

**Changes:** We chose our own rephrasing: "so the full-period LW1 trend is not responsible..."

10 • page 9, line 15, "by year 70": refer to the figure
Comments:
Changes: Done.

• page 9, line 18: why do you use a range of 14-86% here? in other occasions you showed 10-90% percentiles

**15 or 2\*sigma**

**Comments:** We don't reject normality (Kolmogorov-Smirnov, in previously discussed Supplementary Table). I considered the appropriate value here, and went with 2 standard deviations. **Changes:** Changed to mean  $\pm 2$  standard deviations to be more consistent with other approaches.

20 • page 9, line 19-21:

-Could the following maybe also be an explanation: when there is sea ice in NoS, but no sea ice left in SoN, I expect that the cloud radiative effect in SoN is larger because there is more evaporation from the ocean's surface. When later both NoS and SoN are ice-free, the cloud radiative effect (and the downward LW radiations) would be more similar.

25 -Can you diagnose the transition from snow to rain from the model output to confirm your hypothesis?

-Are the radiative properties of rain also considered in your model or are these totally negligible?

**Comments:** We agree this is a solid argument. Exploring this in detail and attempting to partition this quantitatively into e.g. cloud fraction/optical depth/ctP components is tricky, uncertain and distracts from our main points, so we have instead made other changes.

Changes: We added text here and in methods by discussing how our SoN-NoS flux differences include all

5 coupled changes due to inclusion of FIRE. We use your suggestion as an example of such a coupled process. Text added to Section 2.1 saying that rain is excluded, citing Behrangi et al. (2016) which shows that CloudSat R04 products suggest snowfall dominates precip so this probably doesn't matter much.

• page 9, line 25: does your simulated output confirm that the sea ice thickness becomes thinner?

10 Comments: Yes it does. Added analysis combined with the Massonnet et al. discussion is compelling evidence in support of our arugment.

**Changes:** New Methods section 2.3 explains CESM1 thickness analysis and new Figure 4 shows results. References added elsewhere to this evidence, including in discussion/conclusions.

Page 10, lines 13-14, "two models that include FIRE show substantially more summertime SW...":
 -more than what (CMIP5 median)?

-Can you show this somewhere or provide some numbers?

**Comments:**

**Changes:** Pointer to Figure 6(d) added, text changed and example value given.

**20**

• page 11, line 1, "too much surface shortwave radiation": "too much downward shortwave radiation"?

**Comments:**

**Changes:** Changed to SW↓ as throughout.

**25**

• page 11, lines 17-20: Can you calculate from your model output how much sea ice has melted in your simulations (in SoN and NoS)?

**Comments:**

**Changes: Text added based on new Figure 4. Conclusion: ~30 cm difference in mean state for years 1—20.**

• Page 11, lines 21-23: Why did you actually not look at least at some other variables? As an example, it should be easy to see how different the clouds and precipitation are between the two simulations (e.g. liquid

5 water path, cloud cover, snow versus rain).

**Comments:** As stated above, we thought that the necessary justification is supplied by the presented changes in fluxes, sea ice extent and thicknesses. The flux differences alone are, we believe, sufficient to achieve the two functions of our paper: (1) test our main proposed hypothesis and (2) determine whether FIRE can play a large role in simulated Arctic sea ice change.

10 Changes: See previous added text, focus on fluxes and added thickness we feel is sufficient.

• page 11, line 26, "lead to counteracting processes": do you mean: "may disperse the snow radiative effect"? **Comments:** This was meant to highlight how (1) CESM1-CAM5 might have stronger FIRE than other implementations and (2) if other modellers add FIRE, then subsequent tuning of other parameters could us retrieve the angle addresses.

15 counteract the sea ice changes.

Changes: We have rephrased.

• Page 12, line 1, being approximately twice as fast: Do you show that somewhere in the paper? How many years from now on for the two cases?

20 **Comments:** We said "approximately" to give an order of magnitude, this can be seen from figures in the paper.

**Changes:** We have pointed at Figure 2(d).

- Figures in general: Sometimes you use parentheses and sometimes square
- 25 brackets around the units.

**Comments:**

**Changes:** All converted to square brackets for units.

• Figure 1, caption, 10-90% range: I would write "10-90% percentile range" (in the whole text) to be more precise

**Comments:** We think this is precise enough and given that it's used in other papers so should be clear to most readers, we prefer to keep the shorter phrasing.

5 Changes: N/A

• Figure 2, caption: "and" before CESM1-CAM5

**Comments:**

Changes: Caption changed to refer to panel labels a-d.

10

• Figure 3, caption: please delete "but any comparison must be carefully made ...".

In my opinion, statements like this do not belong to a caption but only to the main text.

**Comments:**

Changes: Sentence added in Figure 3 main text discussion.

```
15
```

- Figure 5:
- caption: mention that this figure shows 1pctCO2
- The units should be  $W/m^2$ .

Comments: Thanks for paying so much attention and catching this error.

20 Changes: Done.

• Figure 6, caption: delete "poleward of 30 degree" since you show output between 60 and 90 degree N **Comments:**

**Changes: Done**

**25**

• Supplementary Material, Table 1:

• "whether they exclude falling ice radiative effects": this sounds as if the models have FIRE implemented but exclude them; how about "neglect falling ice radiative effects"?

• "this subset is all those for whom": please rewrite, e.g. "All r1i1p1 simulations were considered that provide the scenarios of interest and the necessary output of surface fluxes and sea ice fields."

**Comments:** "neglect" sounds judgmental to us.

Changes: Rephrased to say whether they "simulate" FIRE or not.

5 Final sentence rephrased to "...that provide the necessary surface flux and sea ice fields for the scenarios of interest."

• Supplementary Figure 1: to what do the colour of the points correspond to (seasons)? If there are more than 8 simulations that you compared, you could add in the caption that the other plots look similar (if this is the

10 case)

**Comments:** Models are those for which we had Dr. Kirchmeier-Young's output for comparison. **Changes:** Caption rephrased to try and better emphasise that colours refer to calendar months.

• Supplementary Figure 3, caption: first you write that the anomaly is relative to 1979-1984, then you write

15 that you calculated the anomalies relative to 1979 (?)

**Comments:**

Changes: This was a typo, we have corrected to 1979–1984 in all cases.

• Supplementary Figure 4, "SIE change is shown as a fraction relative to its 1979-

20 1984 mean": I would rather write that Supplementary Figure 4 shows relative changes (instead of absolute changes).

**Comments:**

Changes: Done.

• Supplementary Figure 5:

• "No uncertainties are shown...": You could detrend the time series before you calculate the standard deviation.

**Comments:**

**Changes:** Done, and we show 2 standard deviations to be consistent with other figures. Caption has been rewritten and points are offset to prevent overlap.

• Supplementary Figure 6, "and may be an underestimate...":

5 –I would not write that in the caption but discuss it in the text.

-Do the lag-1 correlations that you mention refer to individual months? If yes, could you calculate the trend considering the lag-1 correlation for each month individually? Does it make a large difference if you account for autocorrelation? How does it change if you take another trend estimator than linear regression?

-Please mention how you calculate the sigma. Is this the standard deviation of the white noise? Or is it the

10 uncertainty of the trend (which would be more important from my point of view)?

-The error bars overlap for many months and therefore it is impossible to see the standard deviations.

**Comments:** Based on Supplementary Table 2 and main text discussion we switch to white noise only and have checked results with Theil-Sen.

Changes: Discussion of uncertainties and AR(1) removed

15 Caption now describes the sigma calculations, they are uncertainty of the trend. Points have been offset slightly so that the bars can be seen on inspection.

Technical corrections:

20

• Page 1, line 24, "downward shortwave": I would (always in the paper) write "downward shortwave radiation" (the same of course for longwave)

**Comments:** Agreed, but this is used a lot so we introduce  $LW_{\downarrow}$  and  $SW_{\downarrow}$  notation in the Introduction and sometimes use that.

Changes: "Radiation" added to the abstract and introduction, often shorthand thereafter.

Page 2, lines 10-11, "Physically, ice affects both...": Physically, sea ice affects both..."
Comments:
Changes: Change made.

• Page 2, line 13, "From a surface perspective": the previous sentence also refers to the surface **Comments:**

**Changes:** Changed to "Throughout the year..."

5 • Page 2, line 14, "sea-ice extent": "sea ice extent"
Comments:
Changes: Change made.

• page 3, line 1, "From analyses of subsets of climate models in the Climate Model Intercomparison

Project, phase 5 (CMIP5 (Taylor et al., 2012)), ...": This sentence sound complicated. Why not: "Based on CMIP5 data (Climate Model Intercomparison Project, phase 5; Taylor et al., 2012), the observed ..."
 Comments:

**Changes:** Change made. The Cryosphere style in the reference manager won't remove the brackets on the year, but this can be done during editing if accepted.

```
15
```

• page 3, line 7, "are also necessary": "are necessary"

**Comments:**

Changes: Change made.

20 • page 3, line 22, "tends": "tend"

**Comments:**

Changes: Change made

• Page 4, line 5, "increased winter longwave": "increased winter longwave downward radiation"

**25 **Comments:**

**Changes:** We now use  $LW_{\downarrow}$ , having introduced this previously.

• page 4, line 9-10, "This will mean that a non-FIRE simulation should experience more local albedo feedback due to...": "This will mean that a non-FIRE simulation should experience a stronger local snow-albedo feedback due to..."

Comments: This was meant to refer to the sea ice albedo feedback over the ocean, not surface snow. Our

5 argument being that for a given retreat in sea ice cover, the no-FIRE simulation has more SW↓ so a larger dSW/dsic.

Changes: We have changed phrasing to "sea ice albedo feedback".

• page 4, line 16-17, "the direct effect": "the direct consequence"? (because of "radiative effect" in the

10 previous sentence)

**Comments:** This is nicer! **Changes:** Change made.

• page 5, line 3, "who have": "that provide"

15 **Comments:**

Changes: Change made.

• Page 5, line 7, "This is a scenario of very high radiative forcing which we select...": comma before "which"

**20 **Comments:**

Changes: Change made.

• page 5, line 11 (and in general): you use FIRE as a singular but is it not a plural ("falling ice radiative effects")?

25 Comments: Scientific collective acronyms (SCA) is frequently annoying.Changes: We have changed to treat FIRE as plural.

• Page 5, line 12, "and those in which there are no snow radiative effects": "and those in which snow radiative effects are not considered"

**Comments:**

Changes: Change made.

**5**

• page 5, line 13, "These are listed...": "All models are listed..." Comments:

Changes: Change made.

10 • Page 7, line 19: delete "also"

**Comments:**

Changes: Change made.

• Page 8, line 1, "decadal mean SIE": "decadal mean September SIE"

**15 **Comments:**

Changes: Change made.

• page 8, line 7, "potential magnitude": "potential impact"?

**Comments:**

20 Changes: Change made.

• page 8, line 14, "in future models": "in future model versions"?

**Comments:** I feel that "future model versions" implies that current CAM doesn't include it or that future versions might remove it. It is also possible that future models will be developed.

25 Changes: Changed to "in future modelling efforts".

• page 9, line 17: "healthy" sounds colloquial to me

**Comments:** It may be somewhat colloquial but we do not believe that it damages comprehension or reduces precision, so we prefer to keep it. **Changes:** N/A

5 • Page 11, line 12, "shows": showComments:Changes: Change made.

**10**

**Reviewer 2**

Review of Li et al:

**15**

I will keep it short and to the point. Li et al bring in an important aspect into discussion here, i.e. the radiative effects (particularly longwave warming) of falling snow. From the process point of view, I do appreciate that the authors highlight its potential importance and encourage modelling community to take this process into account. The manuscript is written and presented nicely. The analysis is robust and the

20 arguments are justified well based on the results presented here. I do however have few major comments.

The overwhelming focus on the radiative effects, by neglecting the dynamical and surface aspects, concerns me. I understand that the authors neglect them for the sake of simplicity, but they are actually important here. For example, between the two sets of CMIP5 models, SoN and NoS, the former shows

25 more realistic trends in sea-ice extent. Could it be a coincidence? How much of it is really down to including FIRE and/or down to having different dynamical responses and surface descriptions in these

sets of models? Please note that CMIP5 models vary widely in their description of sea-ice (e.g. Koenigk et al., 2014). Could the authors please check how the SoN and NoS models differ in these aspects? FIRE would depend not only on how much it precipitates, but also on the frequency of falling snow. But there seems to be hardly any discussion about this (and how it varies across NoS and SoN). Or am I

5 missing something here?

I hope the authors comment on these issues.

References

10

Koenigk, T., Devasthale, A., and Karlsson, K.-G.: Summer Arctic sea ice albedo in CMIP5 models, Atmos. Chem. Phys., 14, 1987-1998, https://doi.org/10.5194/acp-14-1987-2014, 2014

**Comments:** Thanks for taking the time to read & review our paper. You have highlighted ways in which the original submissions was unclear so we have substantially re-written the paper to address your concerns and those of reviewer 1.

15 Basically, we think you're right: the CMIP5-SoN September retreat looking "better" relative to observations is largely due to chance, so FIRE alone are not a big enough factor to overcome all other inter-model differences. Nevertheless, we are convinced that if the magnitude of FIRE as calculated by CESM1-CAM5 are realistic, then FIRE are important to improve simulated Arctic sea ice.

We don't see the benefit of a detailed analysis of model schemes, beyond how our discussion &

20 conclusions comments that the two CMIP5-SoN models which reach ice free states the earliest are the GISS models and that is likely due to other parts of their cloud schemes resulting in underestimated IWP and way too much summer  $SW_{\downarrow}$ .

We made many, many changes in response to reviewer 1, including extended statistical testing and analysing initial sea ice thickness in CESM1 SoN and NoS, finding that it supports our conclusion. We

25 hope that our changes have clarified our approach and that you agree our main conclusions are suitably supported with the caveats and uncertainties adequately explained.

**Changes:**

Text added to Section 1 details some ways in which CMIP5 sea ice simulations can be affected. We refer to Koenigk et al. (2014) as well as Karlsson & Svensson (2013), and Massonnet et al. (2018) while discussing the variation in sea ice schemes.

5

The bit about mixed phase clouds as an example of how atmospheric components can matter (with the Cesana & Tan papers) has been moved here from the discussion, and a comment on how the representation of ocean eddies with a citation to Horvat & Tziperman (2018) has also been added.

10 Section 5 paragraph 2 now specifically states that "the faster September retreat of CMIP5-SoN in Figure 1 is likely due to the full combination of properties in these models and not directly due to FIRE. Nevertheless, the controlled CESM1-CAM5 simulations demonstrate that the inclusion of FIRE...".

Our introduction now specifies that our two main aims are to determine whether FIRE in simulations 15 leads to important differences in simulated sea ice, and whether our hypothesis of a thinner and more vulnerable pack is supported.

Section 2.1 now includes the following: "The strength of FIRE and the simulated response of other properties to FIRE depend on the frequency as well as the intensity of snowfall. This is accounted for in
the model as radiative transfer is calculated at each model time step even though outputs are only provided monthly." Further text explains that by looking at the differences in radiation terms directly we are actually comparing the fully coupled response due to FIRE, and cites Chen et al. as an example of how FIRE can cause such coupled responses. This means we fully capture the factors that are physically relevant to the sea ice retreat, but we cannot disentangle how much of the "real" cause is direct FIRE
versus changes induced in circulation. We think that our re-written paper is sufficiently careful in its phrasing to emphasise this point and limit our conclusions to those for which we have sufficient supporting evidence.

**Potential faster Arctic sea ice retreat triggered by snowflakes' greenhouse effect**

J-L F. Li1, Mark Richardson1,2, Wei-Liang Lee4, Eric Fetzer1, Graeme Stephens1, Jonathan Jiang1 Yulan Hong3, Yi-Hui Wang6, Jia-Yuh Yu7, Yinghui Liu5

[revised manuscript text omitted]
 Some combination of changes in cloud properties or precipitation phase in response to sea ice cover, 
[revised manuscript text omitted]